# Biogenic isoprene emissions, dry deposition velocity and surface ozone concentration during summer droughts, heatwaves and normal conditions in Southwestern Europe

Antoine Guion[1*], Solène Turquety[1**], Arineh Cholakian[2], Jan Polcher[2], Antoine Ehret[1], and Juliette Lathière[3]

[1]LMD/IPSL, Sorbonne Université, ENS, PSL Université, École polytechnique, Institut Polytechnique de Paris, CNRS, Paris, France.
[*]Now at INERIS, Verneuil-en-Halatte, France
[**]Now at LATMOS/IPSL, Sorbonne Université, UVSQ, CNRS, Paris, France
[2]LMD/IPSL, École polytechnique, Institut Polytechnique de Paris, ENS, PSL Université, Sorbonne Université, CNRS, Palaiseau, France
[3]LSCE/IPSL, CEA-CNRS-UVSQ, Université Paris-Saclay, Gif-sur-Yvette, France

**Correspondence:** Antoine Guion (antoine.guion@ineris.fr)

**Abstract.** At high concentration, tropospheric ozone ($O_3$) deteriorates air quality, inducing adverse effects on human and ecosystem health. Meteorological conditions are key to understand the variability of $O_3$ concentration, especially during extreme weather events. In addition to modifying photochemistry and atmospheric transport, droughts and heatwaves affect the state of vegetation and thus the biosphere-troposphere interactions that control atmospheric chemistry, namely biogenic emissions of precursors and gas dry deposition. A major source of uncertainty and inaccuracy in the simulation of surface $O_3$ during droughts and heatwaves is the poor representation of such interactions. This publication aims at quantifying the isolated and combined impacts of both extremes on biogenic isoprene ($C_5H_8$) emissions, $O_3$ dry deposition and surface $O_3$ in Southwestern Europe.

First, the sensitivity of biogenic $C_5H_8$ emissions, $O_3$ dry deposition and surface $O_3$ to two specific effects of droughts, the decrease in soil moisture and in biomass, is analyzed for the extremely dry summer 2012 using the biogenic emission model MEGANv2.1 and the chemistry-transport model CHIMEREv2020r1. Despite a significant decrease in biogenic $C_5H_8$ emissions and $O_3$ dry deposition velocity, characterized by a large spatial variability, the combined effect on surface $O_3$ concentration remains limited (between +0.5% and +3% over the continent).

The variations of simulated biogenic $C_5H_8$ emissions, $O_3$ dry deposition and surface $O_3$ during the heatwaves and agricultural droughts are then analyzed for summer 2012 (warm and dry), 2013 (warm) and 2014 (relatively wet and cool). We compare the results with large observational data sets: $O_3$ concentrations from Air Quality e-Reporting (2000-2016) and total columns of formaldehyde ($HCHO$, used as a proxy of biogenic emissions of volatile organic compounds) from the Ozone Monitoring Instrument of the Aura satellite (2005-2016).

Based on a cluster approach using the Percentile Limit Anomalies indicator, we find that $C_5H_8$ emissions increase by +33% during heatwaves compared to normal conditions, do not vary significantly during all droughts (accompanied or not by a

heatwave) and decrease by -16% during isolated droughts. OMI data confirm an average increase of $HCHO$ during heatwaves (between +15 and +31% depending on the product used) and decrease of $HCHO$ (between -2 and -6%) during isolated droughts over 2005-2016 summers. Simulated $O_3$ dry deposition velocity decreases by -25% during heatwaves and -35% during all droughts. Simulated $O_3$ concentrations increase by +7% during heatwaves and by +3% during all droughts. Compared to observations, CHIMERE tends to underestimate the daily maximum $O_3$. However, similar sensitivity to droughts and heatwaves are obtained. The analysis of the AQ e-Reporting data set shows an average increase of +14% during heatwaves and +7% during all droughts over 2000-2016 summers (for an average daily concentration value of $69\mu g/m^3$ under normal conditions). This suggests that identifying the presence of combined heatwaves is fundamental to the study of droughts on surface-atmosphere interactions and $O_3$ concentration.

## 1  Introduction

Tropospheric ozone ($O_3$) plays a critical role in maintaining the oxidative capacity of the troposphere. However, as a high oxidant, it also deteriorates air quality at high concentrations, inducing adverse effects on human and ecosystem health. Both short and long-term $O_3$ exposure significantly increase the risk of morbidity and mortality from cardiovascular or respiratory causes (e.g. Jerrett et al., 2009; Nuvolone et al., 2018). It also causes visible damage on plant epidermis as well as photosynthesis inhibition, seriously threatening vegetation growth (Anand et al., 2021) and crop yields (De Andrés et al., 2012). Tai et al. (2014) estimate that $O_3$ pollution enhanced by global warming (Representative Concentration Pathway 8.5) could lead to a global crop yield reduction by 3.6% in 2050.

Tropospheric $O_3$ is a secondary air pollutant formed by photochemical reaction chains initiated by the oxidation of volatile organic compounds ($VOCs$) in the presence of nitrogen oxides ($NO_x$). $VOC/NO_x$ emission ratio determines the chemical regime of $O_3$ production at local scale in a highly nonlinear relationship (Jacob, 1999). Methane ($CH_4$) and carbon monoxide ($CO$) are also $O_3$ precursors in background conditions.

Meteorological conditions are a key driver for $O_3$ formation and transport (Mertens et al., 2020). Indeed, $O_3$ production is favoured by warm and sunny conditions, and $O_3$ peaks therefore occur mainly during summer. The overall objective of this paper is to quantify the variation of surface $O_3$ concentrations over the Euro-Mediterranean area during extreme weather events of heatwaves and droughts. The Euro-Mediterranean region is often affected by both heatwaves (Russo et al., 2015) and droughts (Spinoni et al., 2015). The intensity and frequency of such events have increased over the last decades in the region. According to projections based on climate models, these trends could last and worsen over the 21st century (Perkins-Kirkpatrick and Gibson, 2017; Spinoni et al., 2018).

Heatwaves are often associated with high $O_3$ concentration. For example, Vautard et al. (2005) show that the persistent heatwave of August 2003 led to high $O_3$ concentration over the Western Europe for several days (daily maximum above $150\mu g/m^3$, spatially averaged). Jaén et al. (2021) report hourly values of $O_3$ exceeding $250\mu g/m^3$ during a shorter heatwave (28-29 June 2019) at local scale (Barcelona). These values are well above the European Union (EU) recommendation of a 8h-average maximum of $120\mu g/m^3$.

Porter and Heald (2019) highlight that despite a strong correlation between surface temperature and $O_3$ concentration identified in Europe, this relationship is nonlinear and complex as it depends on several mechanisms including precursor emissions, reaction rates and lifetimes in the atmosphere. High temperatures favour emissions of biogenic $VOCs$ ($BVOCs$), which can induce an increasing formation of $O_3$ in the case of a $NO_x$-limited regime as many areas in the Mediterranean basin (Richards et al., 2013). Persistent and intense solar radiation also allows long photochemical episodes (Jaén et al., 2021). Heatwaves in the Mediterranean being often related to a blocking situation of atmospheric systems, stagnant anticyclonic conditions lead to the accumulation of both precursors and $O_3$ (Otero et al., 2021). Moreover, extreme temperature and high $O_3$ concentration can cause stomatal closure reducing the dry deposition of $O_3$, as highlighted by Gong et al. (2020) with the $O_3$-vegetation feedback.

Droughts also significantly impact atmospheric chemistry through biosphere and atmospheric cycle perturbation (Wang et al., 2017). Drought impact on air pollutants and more specifically $O_3$ is difficult to investigate for several reasons. It is firstly related to their characterization. Due to their multiscalar character, droughts can last on timescales ranging from days to months, making their extent difficult to assess (Vicente-Serrano et al., 2010). Moreover, drought definition depends on the type considered (Svoboda and Fuchs, 2016). Meteorological droughts correspond to a rainfall deficit or an excess of evapotranspiration, agricultural ones to soil water shortage for plant growth (soil dryness), and hydrological ones to surface and/or underground flow decrease. Another difficulty is that droughts affect not only the atmosphere but also the land biosphere through soil dryness and plant activity decline (e.g. Vicente-Serrano, 2007; Vicente-Serrano et al., 2013).

The variability of $O_3$ concentration is generally not well represented in chemistry-transport models (CTMs) compared to observations partially because of the lack of interactions between the meteorology, terrestrial biosphere and atmospheric chemistry (Wang et al., 2017). Uncertainties related to BVOC emission modelling could lead to a BVOC-derived $O_3$ uncertainty estimated at about 50% in Europe (Curci et al., 2010). Another major cause of uncertainties concerns the simulation of meteorological conditions.

Meteorological conditions specific to droughts are low relative humidity, low or absence of precipitation and cloud cover. The latter leads to an increase of downward solar radiation. Temperature can also be higher during droughts than during normal conditions. Wang et al. (2017) found that drought periods in the United States over 1990-2014 were characterized by temperatures up to 2°C hotter and radiation $12.4W/m^2$ higher. Their analysis of surface measurements shows a mean enhancement in $O_3$ concentration of 3.5 ppbv (8%), explained by enhanced precursor emissions and photochemistry. Finally, large amount of $O_3$ precursors emitted during biomass burning enhanced by droughts and heatwaves, can contribute to $O_3$ pollution peaks (e.g. Hodnebrog et al., 2012).

The development of droughts and heatwaves can be linked (Miralles et al., 2019). For example, through the soil moisture-temperature feedback, droughts can increase heatwave intensity due to a decrease in evapotranspiration and an increase in sensible heat (e.g. Zampieri et al., 2009). It is therefore important to integrate such interactions for the simulation of droughts and heatwaves.

Through soil moisture deficit, droughts can induce considerable adverse effects on the biosphere, leading to a decrease of the overall biomass but also of the BVOC emission activity. Guion et al. (2021) report an averaged decrease of $\sim 10\%$ of Leaf Area

Index (LAI) observed by satellite during droughts in Southern Europe. Areas of low altitude and vegetation with short root
systems are more sensitive. Based on simulations from the Model of Emissions of Gases and Aerosols from Nature (MEGAN)
(Guenther et al., 2006, 2012) and using MODIS LAI between 2003 and 2018, Cao et al. (2021) find that vegetation biomass
variability in China is a major controlling factor of the inter-annual variation of BVOC emissions. Furthermore, Emmerson
et al. (2019) simulated (also with MEGAN) a reduction in $C_5H_8$ emissions of 24-52% over Southeastern Australia due to soil
dryness in summer.

Dry deposition velocity in the Mediterranean depends on both stomatal and non-stomatal conductance (Lin et al., 2020; Sun
et al., 2022). Stomatal conductance that can be considerably affected under hot and dry conditions, depends on several param-
eters. As identified by Kavassalis and Murphy (2017), the leaf-to-air vapor pressure deficit that depends on both temperature
and relative humidity, is a key variable regulating the stomatal conductance. Extreme hot and dry conditions can cause stomatal
closure to reduce transpiration. Moreover, lack of rainfall prevents wet deposition and plant water stress slows dry deposition.
However, there are still large uncertainties about the relationship between the soil moisture and the gas dry deposition (e.g.
Clifton et al., 2020b).

Considering the difficulty for models to simulate accurately soil moisture (Cheng et al., 2017) and the uncertainties about plant
and BVOC species response to water stress (Bonn et al., 2019), the impact of soil moisture is often dismissed in biogenic emis-
sions and dry deposition schemes used in CTMs. Therefore, knowledge about the respective effect on $O_3$ remains limited. To
evaluate the effects of droughts and heatwaves on surface $O_3$ in Southwestern Europe, the variation of both BVOC emissions
and dry deposition as a function of meteorological and hydrological conditions are analyzed in this paper including the soil
dryness effect.

Agricultural droughts and heatwaves are identified based on the coupled meteorological and land surface vegetation regional
model RegIPSL for the 1979-2016 period (Guion et al., 2021). Summers are less affected by drought events (33% of them) than
by heatwaves (97%). Nevertheless, drought periods are generally much longer than heatwaves (22.3 days against 5.8). Those
two types of extreme weather events can be considerably correlated over the Western Mediterranean (temporal correlation R
between 0.4 and 0.5).

Atmospheric chemistry is simulated using the regional CHIMERE CTM (Menut et al., 2021) over three selected summers
(2012, 2013 and 2014), including the online calculation of biogenic emissions from the MEGANv2.1 model. Simulations
are analyzed in conjunction with surface $O_3$ observations from the European surface network Air Quality e-Reporting (https:
//www.eea.europa.eu/data-and-maps/data/aqereporting-9). Concentrations of formaldehyde ($HCHO$) are used as a proxy for
BVOC emissions. $HCHO$ is produced in high yield during the oxidation of hydrocarbons. As an intermediate product of the
oxidation of most VOC and being characterized by a short lifetime (about a few hours), $HCHO$ has been used in many studies
to infer VOC emissions (e.g. Millet et al., 2006, 2008; Curci et al., 2010). Observations of $HCHO$ total column by the Ozone
Monitoring Instrument (OMI) onboard the Aura satellite (Levelt et al., 2006) are used.

The observations and models used for this research are presented in Sect. 2 and 3, respectively. The RegIPSL, CHIMERE
and MEGAN models, as well as the different modelling experiments undertaken are detailed.Validation of surface $O_3$, $NO_2$
and 2m temperature are presented in Sect. 4. Results (see Sect. 5) are divided into three parts. Firstly, the sensitivity of $C_5H_8$

emissions, $O_3$ dry deposition and surface $O_3$ concentration to drought effects (biomass decrease and soil dryness) is assessed in

Sect. 5.1. Secondly, the variation of observed and simulated $O_3$, integrating drought and heatwave effects, is analyzed with respect to the variation of troposphere-atmosphere interactions during extreme weather events (see Sect. 5.2). Finally, the number and distribution of days exceeding the EU air quality standard are presented (see Sect. 5.3).

## 2 Observations

### 2.1 European surface network of $O_3$ and $NO_2$ concentration

The in-situ measurements of surface $O_3$ and $NO_2$ provided by the European Environment Agency are used (EEA, https://www.eea.europa.eu/data-and-maps/data/aqereporting-9, last access: 30 July 2021). The AQ e-Reporting project gathers air quality data provided by EEA member countries as well as european collaborating countries. Merging with the statistics of AirBase v8 project (2000-2012), AQ e-Reporting offers a multi-annual time series of air quality measurement data until 2021, categorized by station types and zones across Europe. Only measurements from stations classified as background and rural are

considered for this study.

Thunis et al. (2013) quantify the various sources of uncertainty for $O_3$ measurements (e.g. linear calibration and ultraviolet photometry) and estimated a total uncertainty of 15%, regardless of concentration level. There are also considerable uncertainties in the measurements of $NO_2$. Lamsal et al. (2008) emphasize that the chemiluminescence analyzer, the measurement technique primarily found in air quality stations, is subject to significant interference from other reactive species containing

140 oxidized nitrogen (e.g. PAN, $HNO_3$). This can lead to an overestimation of measured $NO_2$ concentrations.

### 2.2 Satellite observations of $HCHO$ total column

The $HCHO$ total column retrieved from the observations of the backscatter solar radiation in the ultraviolet (306 and 360 nm) by the OMI/Aura instrument are used as a quantitative proxy of BVOC emissions. OMI has an equator crossing time at 01:45 pm on the ascending node, about 14.5 revolutions per day and a swath width of 2600 km, allowing daily global coverage

(Levelt et al., 2006, 2018).

Uncertainties on the $HCHO$ retrieval is significant, varying between 30% (pixels with high concentrations) and 100% (with low concentrations), mainly due to cloud and aerosol scattering along the field of view (González Abad et al., 2015). Satellite retrievals of $HCHO$ have been shown to have systematic low mean bias (20-51%) compared to aircraft observations (Zhu et al., 2016).

For this reason, we chose to focus on the relative difference in total HCHO during droughts and heatwaves (no correction factor has been applied). In order to quantify the uncertainty on the $HCHO$ anomalies obtained, the analysis has been performed using two products. We primarily use the OMHCHOd level 3 product (Chance, 2019), which provides $HCHO$ total column with a spatial resolution of $0.1° \times 0.1°$. For comparison, the level 2 retrieval by the Belgian Institute for Space Aeronomy (BIRA) is also used (De Smedt et al., 2015), thereafter referred to as OMI-BIRA. Both products are included in the intercomparison

conducted by Zhu et al. (2016).

Only observations with a cloud fraction less than 0.3, a solar zenith angle less than 70° and a vertical column density within the range of -0.8x10$^{15}$ and 7.6x10$^{15}$ $mocelules/cm^2$ are selected in order to minimize OMI row anomalies, following Zhu et al. (2017). For a suitable comparison with model simulations, $HCHO$ data were regridded on the Med-CORDEX domain using a bilinear method.

## 2.3 In-situ observations of 2m temperature, $C_5H_8$ surface concentration and $O_3$ dry deposition flux

To complete the validation of our simulations, several observational data sets are added to our study. Firstly, in-situ observations of temperature at 2 meters above the surface ($T_{2m}$) from the E-OBS data set (Cornes et al., 2018) are used to validate the simulated temperature by the Weather Research and Forecasting (WRF) model (Skamarock et al., 2008). The E-OBS gridded product with a resolution of 0.25° × 0.25° was regridded to the Med-CORDEX domain (see Sect. 3.1.1).

Secondly, flux measurements are used for comparison with the different modelling experiments carried out to analyze the sensitivity of biogenic $C_5H_8$ emissions and $O_3$ deposition to the effects of biomass decrease and soil dryness. However, there are few flux measurements available that cover at least several weeks during summer 2012. The ERSA station (FR0033R), located at Cape Corsica in France (42.97°N, 9.38°E) is used to assess surface concentration of $C_5H_8$ measured by a steel canister instrument at 4.0 meters above the surface. The data are provided by the EBAS infrastructure (https://ebas.nilu.no/).

Dry deposition flux measurements of $O_3$ are also used at one station. This is the Castelporziano station (IT-Cp2) located in the Lazio region of Italy (41.70°N, 12.36°E) measuring at 14.9 meters above the surface (eddy covariance technique). A full description of the measurement data is provided in Fares et al. (2013). The data were downloaded from the European Fluxes Database Cluster (EFDC, http://www.europe-fluxdata.eu/).

## 3 Models

## 3.1 RegIPSL (WRF-ORCHIDEE)

### 3.1.1 Model description

The regional WRF-ORCHIDEE model (RegIPSL) couples the land surface model ORCHIDEE (ORganising Carbon and Hydrology In Dynamic EcosystEms) (Maignan et al., 2011) and the meteorological model WRF. ORCHIDEE is composed of three modules; SECHIBA that simulates the water and energy cycle, STOMATE that resolves the processes of the carbon cycle 180 allowing therefore an interactive phenology and finally, LPJ that computes the competition between the Plant Functional Types (PFTs). However, LPJ module is not used for this research.

RegIPSL model has been used following the recommendation of the Med-CORDEX (Coordinated Regional Climate Downscaling Experiment over the Mediterranean domain) initiative (Ruti et al., 2016). On a Lambert-conformal projection, the domain covers the Euro-Mediterranean with a spatial resolution of 20km. The full description of the configuration used with 185 RegIPSL is presented in Guion et al. (2021).

The performed Med-CORDEX simulations over the 1979-2016 period have been evaluated against surface observations of temperature and precipitation, demonstrating that it is suitable for research on droughts and heatwaves (Guion et al., 2021).

### 3.1.2 Indicators of drought and heatwave

Based on simulations performed by the RegIPSL model, we identified heatwaves and agricultural droughts over the Euro-Mediterranean using indicators of extreme weather events, as described in Guion et al. (2021) (data access: https://thredds-x. ipsl.fr/thredds/catalog/HyMeX/medcordex/data/Droughts_Heatwaves_1979_2016/catalog.html).

Following the approach of Lhotka and Kyselý (2015), we computed the Percentile Limit Anomalies of 2m above surface temperature ($PLA_{T2m}$) for heatwave detection and of soil dryness ($PLA_{SD}$) for agricultural drought detection. The soil dryness is computed as the complement of the soil wetness index, which is described in ORCHIDEE as the ratio between the soil moisture and the accesible water content.

For each cell of longitude ($i$) and latitude ($j$) of the domain, the monthly distribution of the 2m temperature/soil dryness is normalized and the $75^{th}$ percentile is computed ($p = 0.75$). The daily $PLA_{T2m/SD}$ indicator is equal to the daily deviation ($dX_{i,j,d}$) between the reference variable ($X_{i,j,d}$) and the corresponding monthly percentile ($X_{i,j,m}^p$) (Equation 1). A heatwave/drought is identified when the daily $PLA_{T2m/SD}$ is positive for three consecutive days.

$$dX_{i,j,d} = X_{i,j,d} - X_{i,j,m}^p \tag{1}$$

The chosen percentile for the detection of extreme weather events is generally higher, between 80 and 95 (e.g. Stéfanon et al., 2012). We chose the percentile 75 in this study for a larger population. As a result, events detected here cover not only extreme events, but also periods that are significantly drier and warmer than normal conditions.

$PLA_{T2m}$ was calculated based on 2m temperature observations (E-OBS data set) in Guion et al. (2021). Although the intensity of heatwaves was slightly overestimated with the Med-CORDEX simulations ($+0.16°C$ for the mean bias and $+0.50°C$ for the maximum bias), their temporal correlation with observations was high (R coefficient of about 0.9 over the whole Mediterranean).

### 3.1.3 Soil water stress for plants

The Soil Water Stress ($SWS$) calculated by RegIPSL is used to describe the water stress for plant transpiration and photosynthesis (de Rosnay et al., 2002). It is an index varying between 0 (plants fully stressed) and 1 (not stressed) that takes into account the water transfer between the soil layers along root profiles (11 layers in the version of ORCHIDEE used here). For each PFT ($p$), soil class ($s$) and soil layer ($v$), it is computed as follows:

$$SWS_{p,s} = \frac{SM_{p,s,v} - SMw_{p,s,v}}{SMnostress_{p,s,v} - SMw_{p,s,v}} \times nroot_{p,s,v} \tag{2}$$

where $SM$ is the soil moisture (liquid phase), $SMw$ the level at wilting point, $SMnostress$ the level at which $SWS$ reaches 1 and $nroot$ the normalized root length fraction in each soil layer (between 0 and 1). There are three soil classes to avoid competition for water uptake: bare soil, soil with short root systems and soil with long ones.

## 3.2 CHIMERE

### 3.2.1 Model description and configuration

The CHIMERE v2020r1 Eulerian 3-dimensional regional CTM (Menut et al., 2021) computes gas-phase chemistry, aerosol formation, transport and deposition. It can be guided by pre-calculated meteorology (offline simulation) or used in online with the WRF regional meteorological model (Skamarock et al., 2008), including the direct and indirect effects of aerosols in option. In this study, we chose to use the online version of CHIMERE but without any aerosol effects in order to reduce the calculation time and to isolate possible feedbacks.

Here, the reduced scheme MELCHIOR2 (Derognat, 2003) is used for gas phase chemistry (44 species, almost 120 reactions). The aerosol module includes aerosol microphysics, secondary aerosol formation mechanisms, aerosol thermodynamics and deposition, as detailed in Couvidat et al. (2018); Menut et al. (2021). The photolysis rates are calculated online using the Fast-JX module version 7.0b (Bian and Prather, 2002) which accounts for the radiative impact of aerosols.

The WRF model used with CHIMERE, which has a different configuration from the RegIPSL model (Guion et al., 2021), has 15 vertical layers, from 998hPa up to 300hPa. The physics interface for the surface is provided by the Noah Land Surface Model. The aerosol-aware Thompson scheme (Thompson and Eidhammer, 2014) is used for the mycrophysics parameterization. Horizontal and vertical advection are based on the scheme of Van Leer (1977). The reanalysis meteorological data for the initial and boundary conditions are provided by the National Centers of Environmental Prediction (NCEP). The physical and chemical time steps are respectively 30 and 10 minutes.

The simulations have been performed for June-July-August (with a 5 days spin-up) of years 2012-2013-2014 over two nested simulation domains (Fig. 1). The large domain covers Northern Africa and Europe at 60km horizontal resolution (EUMED60, 164x120 cells) and the small domain is close to that of Med-CORDEX although smaller; it covers the Western Mediterranean region at 20km horizontal resolution (Med-CORDEX, 222x93 cells). The area studied in more detail in the following is the Southwestern Europe (35-46°N, 10°W-20°E).

Chemical boundary conditions for the larger domain are provided by a climatology from the global CTM LMDZ4-INCA3 (Hauglustaine et al., 2014) for trace gases and non-dust aerosols, and from the GOCART model (Chin et al., 2002) for dust. The biogenic emissions are calculated online by the MEGAN model (see Sect. 3.2.2). Sea salts and dimethyl sulfide marine emissions are calculated online using the scheme of Monahan (1986) and Liss and Mervilat (1986), respectively. Mineral dust emissions are also calculated on-line (Marticorena and Bergametti, 1995; Alfaro and Gomes, 2001). Consistently with the WRF model, the land cover classification used for the calculation of biogenic and dust emissions is from the United States Geological Survey (USGS) land cover database (http://www.usgs.gov). The biomass burning emissions from the APIFLAME v2.0 model (Turquety et al., 2020) are used. The anthropogenic surface emissions are derived from the EMEP database at

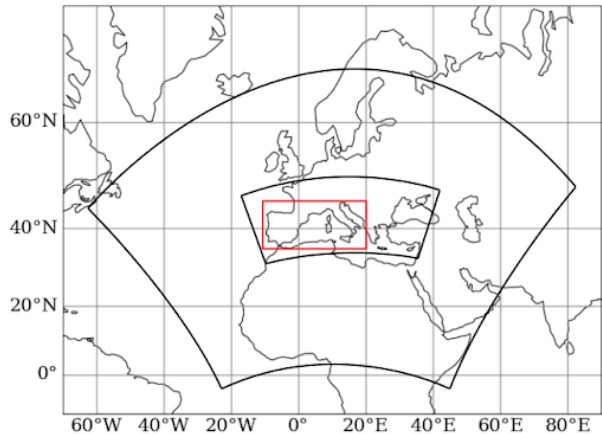

**Figure 1.** Domains used for the CHIMERE simulations: EUMED60 at 60km resolution is the large domain, the smaller nested domain is Med-CORDEX at 20km resolution. The Southwestern Europe region studied here is shown in red.

0.1°×0.1° resolution (https://www.ceip.at/webdab-emission-database) for the year 2014. The inventory was preprocessed by the emiSURF program in order to calculate hourly emissions fluxes on the CHIMERE grid and the period of simulations (https://www.lmd.polytechnique.fr/chimere/formation_2021/emisurf.pdf).

### 250  3.2.2  MEGAN and the soil moisture factor

BVOC emissions were computed using the MEGAN model v2.04 (Guenther et al., 2006), including several updates from the version 2.1 (Guenther et al., 2012). It is used on-line in the CHIMERE model at hourly time step. Emission fluxes are calculated based on emission factors at canopy standard conditions that are defined as $303K$ for the air temperature ($T$), $1500 \, \mu mol/m^2.s$ for the photosynthetic photon flux density ($PPFD$) at the top of the canopy, $5 \, m^2/m^2$ for the LAI and a canopy composed of 255  80% mature, 10% growing and 10% old foliage. Environmental conditions are taken into account using activity factors ($\gamma$) that represent deviations from the canopy standard conditions, so that the emission rate ($ER \, [\mu g.m^{-2}.h^{-1}]$) for a given species $i$ is calculated in each model grid cell as:

$$ER_i = EF_i \cdot \gamma_{LAI} \cdot \gamma_{PPFD,i} \cdot \gamma_{T,i} \cdot \gamma_{Age,i} \cdot \gamma_{SM,i} \cdot \gamma_{CO2,i} \cdot \rho_i \tag{3}$$

where $EF$ is the emission factor $[\mu g.m^{-2}.h^{-1}]$ provided by the MEGAN model and $\rho$ the production/loss term within canopy 260  that is assumed to be equal to 1.

$\gamma_{LAI}$ is the activity factor based on LAI observations from the MODIS MOD15A2H product (Myneni et al., 2015) improved by Yuan et al. (2011) (http://globalchange.bnu.edu.cn/research/lai). This improvement is undertaken with a two-step integrated method: (1) the Modified Temporal Spatial Filter is used to fill the gaps and replace low quality data by consistent ones; (2) the TIMESAT Salvitzky and Golay filter is applied to smooth the final product. The temporal resolution is 8 days.

$\gamma_{PPFD}$, $\gamma_T$ and $\gamma_{Age}$ are the activity factors accounting for light, temperature and leaf age respectively. They are calculated in MEGAN using meteorological variables from the WRF model. The activity factors accounting for the soil moisture ($\gamma_{SM}$) and $CO_2$ concentration ($\gamma_{CO2}$) are applied on $C_5H_8$ species only. $\gamma_{CO2}$ is fixed ($CO_2$ concentration at 395 ppm) and $\gamma_{SM}$ depends on the deviation between the soil wetness and a fixed wilting point, computed as follows:

$$
\begin{cases}
\gamma_{SM} = 1 & \text{if } \theta > \theta_1 \\
\gamma_{SM} = \frac{(\theta - \theta_w)}{\Delta\theta_1} & \text{if } \theta_w < \theta < \theta_1 \\
\gamma_{SM} = 0 & \text{if } \theta < \theta_w
\end{cases}
\tag{4}
$$

where $\theta$ is the soil wetness (volumetric water content in $m^3/m^3$), $\theta_w$ the wilting point (level from which plants can not extract water from the soil, in $m^3/m^3$), $\Delta\theta_1$ equal to 0.04 (empirical parameter) and $\theta_1$ the sum of $\theta_w$ and $\Delta\theta_1$. $\theta$ values (third layer, 60cm depth) are provided by the WRF-Noah land surface model (Ek et al., 2003; Greve et al., 2013). $\theta_w$ values are computed over the domain (Supplementary Information: Fig. S1) using tabulated values from Chen and Dudhia (2001) that are soil type specific and parameterized on Noah soil wetness values. These values are spatialized using the soil texture map provided by the

USGS (STATSGO-FAO product). However, emission response to drought is very sensitive to the wilting point (Müller et al., 2008). Wang et al. (2021b) computed a relative difference of biogenic emissions varying between 50 and 90% depending on the wilting point values.

Based on a comprehensive data set of in-situ measurements of $C_5H_8$ emissions, Bonn et al. (2019) computed an activity factor of soil moisture in function of the soil water availability. The fitted function is relatively in good agreement with the $\gamma_{SM}$

algorithm in MEGAN.

As an alternative, Jiang et al. (2018) propose to use an activity factor of soil moisture that integrates the soil water stress for plant photosynthesis. Below a critical value (fixed here at 0.5), the activity factor using SWS from RegIPSL ($\gamma_{SWS}$) decreases lineary as follows:

$$
\begin{cases}
\gamma_{SWS} = 2\,SWS & \text{if } SWS \leq 0.5 \\
\gamma_{SWS} = 1 & \text{if } SWS > 0.5
\end{cases}
\tag{5}
$$

The different soil wetness functions presented here will be subject to dedicated simulations for comparison.

Soil $NO$ emissions are also included in the different simulations. Emission factors are based on the European soil emission inventory (Stohl et al., 1996) which includes both contribution from forests and agricultural soils. Due to the strong temperature dependence, activity factors of soil $NO$ are processed as for biogenic emissions in the MEGAN model. However, no dependence on soil moisture was parameterized.

### 3.2.3 Dry deposition scheme

Dry deposition in CHIMERE is resolved online using a canopy resistance approach, based on the scheme of the EMEP model (Wesely, 1989; Emberson et al., 2000; Simpson et al., 2003, 2012). Canopy resistance is calculated from stomatal conductance ($g_{sto}$) which increases proportionally with LAI, and the bulk non-stomatal conductance ($G_{ns}$). $g_{sto}$ varies between a maximum ($g_{max}$) and minimum ($g_{min}$) daytime stomatal conductance [mmol $O_3/m^2.s$] depending on a series of meteorological conditions represented by factors of relative conductance ($f$) (Eq. 6). The parameters used to calculate these factors vary with land cover ($lc$) and the seasonality. Because the land cover classes used in the dry deposition scheme are different from the one used in CHIMERE (USGS), a matrix of conversion is applied.

$$g_{sto} = g_{max,lc} \cdot f_{phen,lc} \cdot f_{light,lc} \cdot max(g_{min,lc}, (f_{temp,lc} \cdot f_{VPD,lc} \cdot f_{SWS,lc})) \tag{6}$$

where $f_X$ are factors of relative conductance determined by the leaf/needle age ($X = phen$), irradiance ($X = light$), temperature ($X = temp$), leaf-to-air vapour pressure deficit ($X = VPD$) and the soil moisture ($X = SWS$). Environmental variables required for $f_X$ factors are from the WRF model except for $f_{SWS}$ that is based on SWS from RegIPSL. $f_{SWS}$ for which the calculation is the same as $\gamma_{SWS}$, follows the parameterization suggested by Simpson et al. (2012) for the stomatal conductance. $G_{ns}$ depends on three resistances: the external leaf uptake, the ground surface and the in-canopy. The external leaf uptake resistance is fixed at 2500 $s/m$, the ground surface resistance is estimated from tabulated values (PFT specific) and the in-canopy resistance varies with the surface area index which is expressed in terms of LAI. The LAI used in the dry deposition scheme is parameterized (with no inter-annual variation). Information on phenology and biomass variation for each land cover type was collected from several studies in Emberson et al. (2000).

### 3.3 Modelling experiments

The years 2012-2013-2014 were selected for the large diversity of hydro-climatic conditions and the availability of $O_3$ observations over the study area. During summer 2012, Southwestern Europe was affected by severe droughts ($PLA_{SD} > 0.1$ of soil dryness) and heatwaves ($PLA_{T2m} > 1.5°C$) (SI: Fig. S2). The summer 2013 was wet in the Iberian Peninsula, while the summer 2014 was wet in Italy and the Balkans. A few heatwaves occurred in summer 2013 over Southern France, Northern Italy and Northern Spain. Summer 2014 was colder ($PLA_{T2m}$ of -4.0°C) for the entire study area. The analysis method we propose here is divided into two steps.

Firstly, a focus is made on drought effects (biomass decrease and soil dryness) that are generally poorly represented in modelling experiments, over summer 2012. Several experiments are conducted on the Med-CORDEX domain in order to analyze the sensitivity of simulated $C_5H_8$ emissions, $O_3$ dry deposition and surface $O_3$ concentration to drought effects. Table 1 summarizes the CHIMERE simulations conducted. The "Reference" simulations include all the emissions as defined in Sect. 3.2.1 but without accounting for the soil moisture factor in the emission scheme of MEGAN and deposition one of CHIMERE. A simulation with no biogenic emissions is also performed ("NoBio-emiss") to estimate their relative contribution to $O_3$ concentration.

The LAI used in the emission scheme of MEGAN is year dependent (MODIS observations). To evaluate the effect of biomass decrease by droughts, a simulation with LAI corresponding to a wet summer was used ("HighLAI-emiss"). Summer 2012 which was affected by an important biomass decrease over most of the study area (SI: Fig. S3), has been simulated with the LAI of the wet summer of 2014 (higher than the 2012-2014 mean).

The LAI database used in the dry deposition scheme of CHIMERE does not vary with the year, so that the biomass decrease during the summer 2012 is not reflected. In order to evaluate the importance of this effect, a simulation ("LAIdecr-dep") with a LAI decrease of 5% for forests and 20% for grass and crops (close to the variations observed in 2012 (SI: Fig. S3)) has been conducted.

The effect of soil dryness for the BVOC emissions is evaluated using three different approaches:

- using $\gamma_{SM}$ from WRF-Noah in MEGAN ("$\gamma_{SM}$-emiss" experiment),

- using $\gamma_{SWS}$ from RegIPSL in MEGAN ("$\gamma_{SWS}$-emiss" experiment),

- using $\gamma_{SWSfit}$ from RegIPSL in MEGAN ("$\gamma_{SWSfit}$-emiss" experiment). The fitted function of Bonn et al. (2019) is applied on the SWS as follows:

$$\gamma_{SWSfit} = exp(-exp(0.056 \cdot exp(1) \cdot (-2.3 - SWS) + 1)) \tag{7}$$

The soil dryness effect on the dry deposition is evaluated using the $f_{SWS}$ from RegIPSL in CHIMERE ("$f_{SWS}$-dep" experiment).

Secondly, based on a simulation configuration integrating all drought and heatwave effects ("$\gamma_{SWSfit}$-emiss & LAIdecr/$f_{SWS}$-dep" experiment, hereafter called "all-emiss-dep" experiment), the variation of $C_5H_8$ emissions, $O_3$ dry deposition velocity and surface $O_3$ concentration is analyzed over summers 2012, 2013 and 2014. In order to compare isolated and combined extreme events with normal conditions, we used a cluster approach based on the $PLA_{T2m}$ and $PLA_{SD}$ indicators.

## 4   Validation of surface $O_3$, $NO_2$ and $T_{2m}$

Simulated surface $O_3$ and $NO_2$ concentrations by CHIMERE were compared to the EEA observations while simulated $T_{2m}$ by the WRF model was compared to the E-OBS observations (Cornes et al., 2018). Table 2 presents the mean validation scores of the "Reference" simulations over Southwestern Europe for summer 2012. Validation scores of $O_3$ and $NO_2$ are very similar for summers 2013 and 2014.

The simulated surface $O_3$ is slightly higher than the observations for hourly and daily mean values (bias "observation - model" of -1.93 and -0.13$\mu g/m^3$) and lower for the daily maximum values (+10.94$\mu g/m^3$). The simulated diurnal cycle is lower than the observed one: CHIMERE overestimates the daily minimum and underestimates the maximum. The temporal correlation (R pearson coefficient) varies between 0.53 in daily mean and 0.57 in hourly mean. The spatial distribution of bias and correlation scores of daily maximum values for each summer is shown in SI (Fig. S4). The highest bias ($> +25\mu g/m^3$) and lowest correlations (R of about 0.2) are obtained near large urban areas (e.g. Madrid, Milano) which may be less well represented in

**Table 1.** Name, description and aim of the simulations launched with the CHIMERE model on the nested Med-CORDEX domain for summers 2012, 2013 and 2014.

| Simulation name | Description | Aim |
| --- | --- | --- |
| Reference (R.) | CHIMERE reference (v2020r1) | Default for dry and hot periods |
| *On biogenic emissions* | | |
| NoBio-emiss | R. without biogenic emissions | Contribution of biogenic emissions to $O_3$ |
| HighLAI-emiss | R. with wet summer LAI | Effect of biomass decrease |
| $\gamma_{SM}$-emiss | R. with $\gamma_{SM}$ factor from Noah | Effect of soil dryness |
| $\gamma_{SWS}$-emiss | R. with $\gamma_{SWS}$ factor from ORCHIDEE | Effect of soil dryness |
| $\gamma_{SWSfit}$-emiss | R. with $\gamma_{SWSfit}$ factor from ORCHIDEE | Effect of soil dryness |
| *On gas dry deposition* | | |
| LAIdecr-dep | R. with prescribed LAI reduction | Effect of biomass decrease |
| LAIdecr/$f_{SWS}$-dep | R. with prescribed LAI reduction and $f_{SWS}$ factor from ORCHIDEE | Effect of biomass decrease and soil dryness |

the model even if the stations are classified as rural background. There are substantial uncertainties related to the $NO_x/VOC$ emissions in (peri-)urban areas.

These $O_3$ scores show an overall agreement with those calculated by Gaubert et al. (2014) and Menut et al. (2021) over Europe. The averaged daily maximum bias and correlation coefficients computed are slightly higher and lower, respectively, in this study mainly because the area is limited to Southwestern Europe. Due to the multiple sources of $O_3$ precursors and favorable temperature and light conditions, the Mediterranean is identified as a region of important uncertainty for modelling $O_3$ concentration (Richards et al., 2013).

Validation scores are also computed for the surface $NO_2$ using the EEA observations, with a different distribution of stations that for $O_3$. In agreement with the results of Menut et al. (2021), CHIMERE underestimates $NO_2$ concentrations (+4.95$\mu g/m^3$ in daily mean and +8.57$\mu g/m^3$ in daily maximum) compared to EEA observations. Interference from other oxidized nitrogen species in the chemiluminescence analyzer (e.g. Lamsal et al., 2008) may explain the constant underestimation compared to observations. The mean temporal correlation coefficients do not exceed 0.4. These low validation scores may also be linked

to the emission inventory used as well as meteorological conditions (boundary layer height representation in particular). As a major $O_3$ precursor, part of the uncertainty of this study is directly related to $NO_2$ emissions and concentrations.

Finally, as a critical variable for simulating diurnal and seasonal $O_3$ concentrations, $T_{2m}$ values have been compared to the E-OBS observations. Averaged over the Southwestern Europe, the bias is close to $0^\circ C$ while the RMSE is significant (8.19$^\circ C$ in

**Table 2.** Comparisons between observed and simulated surface $O_3$ concentration, $NO_2$ concentration and 2m temperature, averaged over Southwestern Europe during summer 2012, for the "Reference" CHIMERE simulation.

|  | Observations | Model | Bias (obs. - mod.) | RMSE | Pearson correl. (R) |
|---|---|---|---|---|---|
| $O_3$ | $[\mu g/m^3]$ | $[\mu g/m^3]$ | $[\mu g/m^3]$ | $[\mu g/m^3]$ |  |
| Hourly | 83.21 | 85.14 | -1.93 | 20.45 | 0.57 |
| Daily mean | 83.93 | 84.06 | -0.13 | 14.31 | 0.53 |
| Daily max | 116.32 | 105.39 | 10.94 | 21.49 | 0.54 |
| $NO_2$ | $[\mu g/m^3]$ | $[\mu g/m^3]$ | $[\mu g/m^3]$ | $[\mu g/m^3]$ |  |
| Hourly | 7.65 | 3.70 | 3.95 | 5.57 | 0.25 |
| Daily mean | 7.55 | 2.60 | 4.95 | 5.24 | 0.40 |
| Daily max | 14.65 | 6.08 | 8.57 | 9.87 | 0.37 |
| $T_{2m}$ | $[^\circ C]$ | $[^\circ C]$ | $[^\circ C]$ | $[^\circ C]$ |  |
| Daily mean | 22.25 | 22.25 | -0.01 | 8.19 | 0.76 |
| Daily max | 23.13 | 28.34 | 0.56 | 12.63 | 0.75 |

daily mean and $12.63^\circ C$ in daily maximum). The spatial distribution of the bias presents large variations (SI: Fig. S5). The daily maximum $T_{2m}$ is overestimated in the Southern Mediterranean (up to $5^\circ C$) compared to observations while it is underestimated in the northern part (up to $5^\circ C$). Nevertheless, the averaged scores of temporal correlation are high (around 0.75). Even if such validation scores are close to those found in the scientific literature (e.g. Panthou et al., 2018), the temperature uncertainties significantly contribute to those of the $O_3$ simulated by CHIMERE.

## 5 Results

### 5.1 Sensitivity to soil dryness and biomass decrease effects

The sensitivity analysis to soil dryness and biomass decrease effects is focused on summer 2012 as droughts mainly occurred during this year (over the 2012-2014 period). Most of Southwestern Europe was affected by a biomass decrease in 2012 (compared to the wet summer 2014), except Eastern Spain. The LAI difference between summer 2012 and 2014 can reach -30% over central Italy (Appenines region) corresponding to a mean decrease of 0.5 $m^2/m^2$ (SI: Fig. S3).

$C_5H_8$ is the main contributor to the total mass of $BVOCs$ emitted (70%) in our study area, followed by the model species APINEN (13%, including e.g. $\alpha$-pinene, sabinene).

### 5.1.1 $C_5H_8$ emissions

Figure 2a shows the daily mean $C_5H_8$ emissions in Southwestern Europe for summer 2012 simulated by the MEGAN model ("Reference" simulation). The spatial distribution of $C_5H_8$ over the Mediterranean is heterogeneous, ranging from areas of zero or low emissions ($\sim 1 \times 10^{-3} g.m^{-2}.h^{-1}$) to high emitting areas ($\sim 1.5 \times 10^{-3} g.m^{-2}.h^{-1}$) such as in the Balkans, Apennins, Sierra Morena, Sardinia and Central Portugal. Similar spatial distributions were found for APINEN model species. Differences in spatial patterns among species depend on the variation of emission factors over the land cover classes.

The effect of biomass decrease and soil dryness on $C_5H_8$ emissions during the dry summer 2012 is also shown in Figure 2. Biomass decrease averaged over June, July and August is characterized by negative differences of emissions over most of the southern part of the region, reaching -20% in Northern Spain and -25% in Northern Italy.

The impact of soil dryness is assessed with the "$\gamma_{SM}$-emiss", "$\gamma_{SWS}$-emiss" and "$\gamma_{SWSfit}$-emiss" MEGAN simulations. The quantified impact can vary considerably between them. In the "$\gamma_{SM}$-emiss" experience, $C_5H_8$ emissions stay constant as long as the soil moisture is above the wilting point ($\theta_w$). Once this point is reached (water is missing), it is assumed that plant can not synthesize $C_5H_8$ anymore and the emissions decrease steeply. $\theta_w$ is constant and depends only on the soil type. For instance, silt soil is characterized by lower $\theta_w$ than clay soil, so that soil dryness is more quickly reached for the latter. Due to the high spatial variability of soil type, the difference induced by "$\gamma_{SM}$-emiss" is also characterized by large variability. Within a same region (e.g. Central Italy), the difference sharply varies between -50% and 0% while the region is all over affected by an agricultural drought according to the $PLA_{SD}$ indicator (SI: Fig. S2).

For the alternative approach based on the dynamical $SWS$ function, the relative difference of $C_5H_8$ emissions ("Reference" .vs. "$\gamma_{SWS}$-emiss" experiment) is more spatially homogeneous than "$\gamma_{SM}$-emiss", and the overall reduction is larger. The strongest stress values are located in plains and for plants with short-root systems, in agreement with the sensitivity analysis performed by Vicente-Serrano (2007) about drought effects on vegetation. Semi-arid regions (e.g. Andalusia) are strongly affected as the water budget is almost permanently in deficit (high solar radiation and low/no precipitation). Being adapted to recurrent droughts, some plant species (e.g. *Arundo donax* in a Moroccan ecotype) can reduce the $C_5H_8$ synthesis as the result of pressure selection to preserve their viability (Haworth et al., 2017). However, the emission reduction could be overestimated as irrigation is largely used in many semi-arid areas (e.g. García-Vila et al., 2008). A specific analysis should be undertaken to cover the diversity of responses to drought stress in both natural and human influenced systems within such regions.

The third experiment "$\gamma_{SWSfit}$-emiss" presents the same areas of $C_5H_8$ decrease as "$\gamma_{SWS}$-emiss" but in a lower extent. $C_5H_8$ emissions decrease when SWS values are the lowest. Semi-arid regions are also more affected (-50%) than others (e.g. -20% in Central Italy).

Three areas of high emissions are analyzed more specifically (boxes in Fig. 2): the Balkans, the Pô Valley and Central Spain. Their temporal evolution of $C_5H_8$ emissions during the summer 2012 for the different experiments are shown in Figure 3. Droughts and heatwaves occured over the three regions. Droughts do not negate the dependence on light and/or temperature, and emission peaks are driven by heatwaves with higher 2m temperature and solar radiation (SI: Fig. S6). However, the peak values are reduced if LAI decrease and soil dryness are accounted for. Averaged over the three regions, the biomass decrease

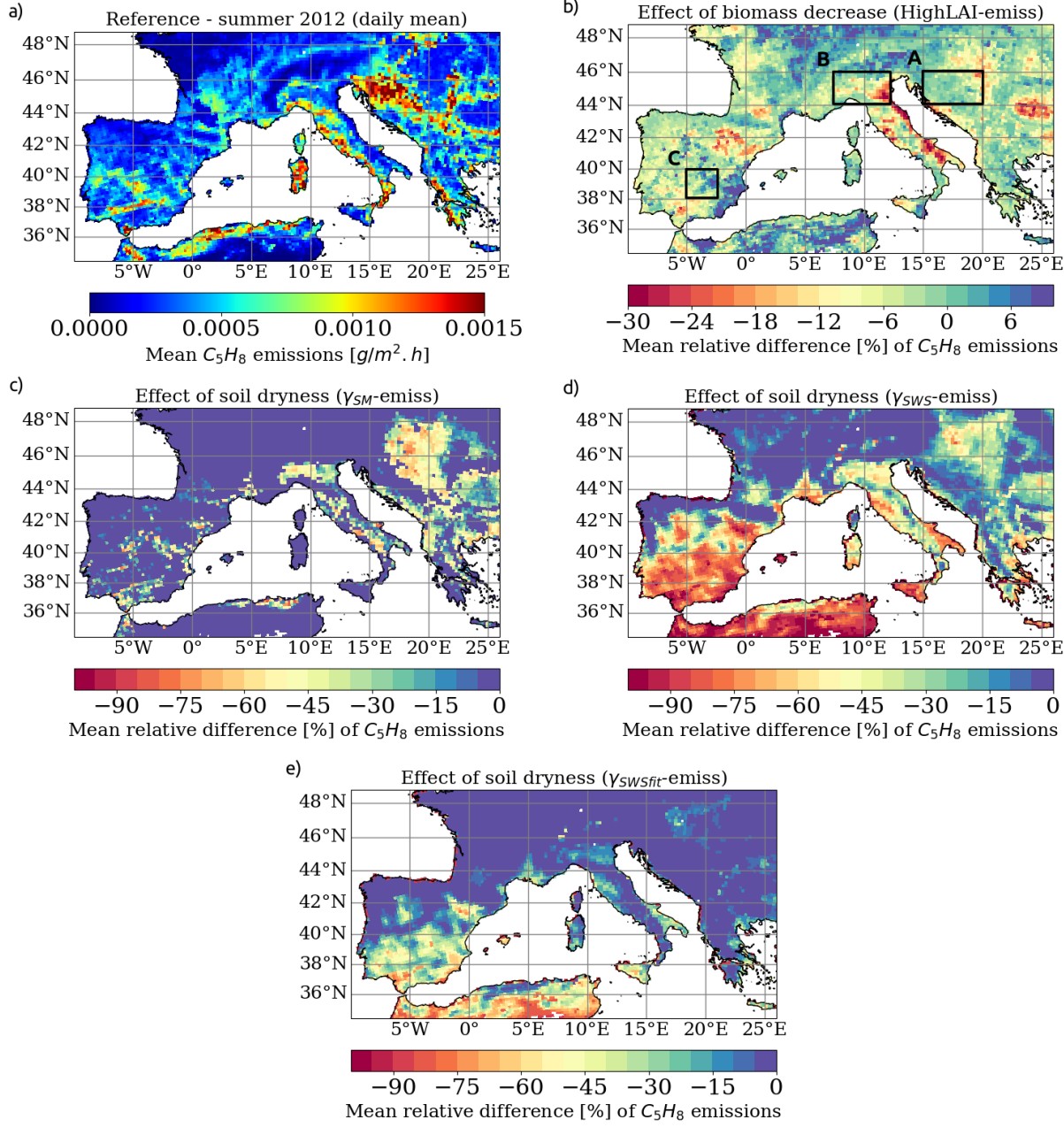

**Figure 2.** Summer mean (JJA) of daily mean $C_5H_8$ emissions $[g.m^{-2}.h^{-1}]$ for 2012 from the "Reference" simulation (a). Mean relative difference [%] of $C_5H_8$ emissions due to biomass decrease and soil dryness. The relative difference is computed between the "Reference" and "HighLAI-emiss" simulation to quantify the biomass decrease effect (b), and "$\gamma_{SM}$-emiss" (c) / "$\gamma_{SWS}$-emiss" (d) / "$\gamma_{SWSfit}$-emiss" (e) to quantify the soil dryness effect. For diagnostic purposes, areas of interest are designated within the dashed rectangles: here named as the "Balkans" (A), "Pô Valley" (B) and "Central Spain" (C).

induces a 3% decrease of the total summer amount of $C_5H_8$ emitted. The soil dryness parameter induces difference of -12% for "$\gamma_{SM}$-emiss", -39% for "$\gamma_{SWS}$-emiss" and -13% for "$\gamma_{SWSfit}$-emiss". Based on experimental measurements over three summers (2012, 2013 and 2014) in a Mediterranean environment (Observatoire de Haute Provence, *Quercus pubescens* plant species), Saunier et al. (2017) identified a 35% emission decrease due to severe droughts. Demetillo et al. (2019) measured in California a $C_5H_8$ concentration reduction of 50% during severe droughts (2014 and 2015). These values are close to and even above the range of the simulated experiments. "$\gamma_{SWS}$-emiss" experiment is considered here as the upper limit of the reduction range of $C_5H_8$ emissions due to dry conditions, $SWS$ being included in the calculation of the emissions rate as soon as soil water stress for plant stress is below 0.5.

In addition, the different simulated experiments from the MEGAN-CHIMERE model have been compared to observations of surface $C_5H_8$ concentrations from EBAS data set (Fig. 4). The Ersa station having an almost complete time series for summer 2012, is kept. Based on the $PLA$ indicator, Northern Corsica was affected by two heatwaves but no drought. However, we detected soil dryness conditions close to the drought limit (mean $PLA_{SD}$ of -0.09). The temporal correlation between simulated and observed $C_5H_8$ concentrations is similar for each experiment (R coefficient around 0.68). Averaged over the summer, the lowest mean bias with the observations is obtained with "$\gamma_{SWSfit}$-emiss" (-28 pmol/mol) and the largest one with "$\gamma_{SM}$-emiss" (-151 pmol/mol). Over July and August, "$\gamma_{SWS}$-emiss" experiment presents the lowest mean bias (+56 pmol/mol). The "Reference" and "$\gamma_{SM}$-emiss" experiments have equal values as the soil wetness from WRF-Noah is above the local wilting point. However, this comparison with observations is made at a single station. It would be valuable to carry out the same exercise at several locations (including areas representative of semi-arid environment) and over several summers. Finally, biogenic emission models (such as MEGAN) consider in general a reduction in emissions when drought episodes occur only for isoprene ($C_5H_8$). However, there is a lot of uncertainty on how plant activity reacts to water stress. In the case of $C_5H_8$ species, the response to water stress could occur in two phases: a state of increasing emissions due to leaf temperature stimulation during the drought onset, followed by a state of $C_5H_8$ synthase limitation (Potosnak et al., 2014). For monoterpenes, the response to water stress may be different depending on the species (Bonn et al., 2019). For instance, sabinene emissions (experimentally measured) from the European beech (*Fagus sylvatica*) decrease strongly with decreasing soil water availability while trans-$\beta$-ocimene emissions (from the same plant species) remain constant.

### 5.1.2 $O_3$ dry depositon velocity

Figure 5a presents the daily mean dry deposition velocity ranging from 1.0 to 1.8cm/s within our areas of interest. The mean effect of biomass decrease (relative difference between the "Reference" and the "LAIdecr-dep" experiment, Fig. 5b) over summer 2012 does not exceed -8%, and is lower over the forested areas (as it was prescribed in our experiment).

The cumulative effects of biomass decrease and soil dryness are included in the "LAIdecr-dep/$f_{SWS}$-dep" experiment (Fig. 5c). The soil dryness effect is much larger than that of biomass decrease. Values range from 0 to -35%. Central Italy and the Iberian Peninsula are the most affected regions.

The temporal evolution of $O_3$ dry deposition velocity for the three areas of interest is shown in Figure 6. The biomass decrease effect is larger (e.g. -9% over the Pô Valley) during the first half of summer. It is close to -1% at the end of summer. This is

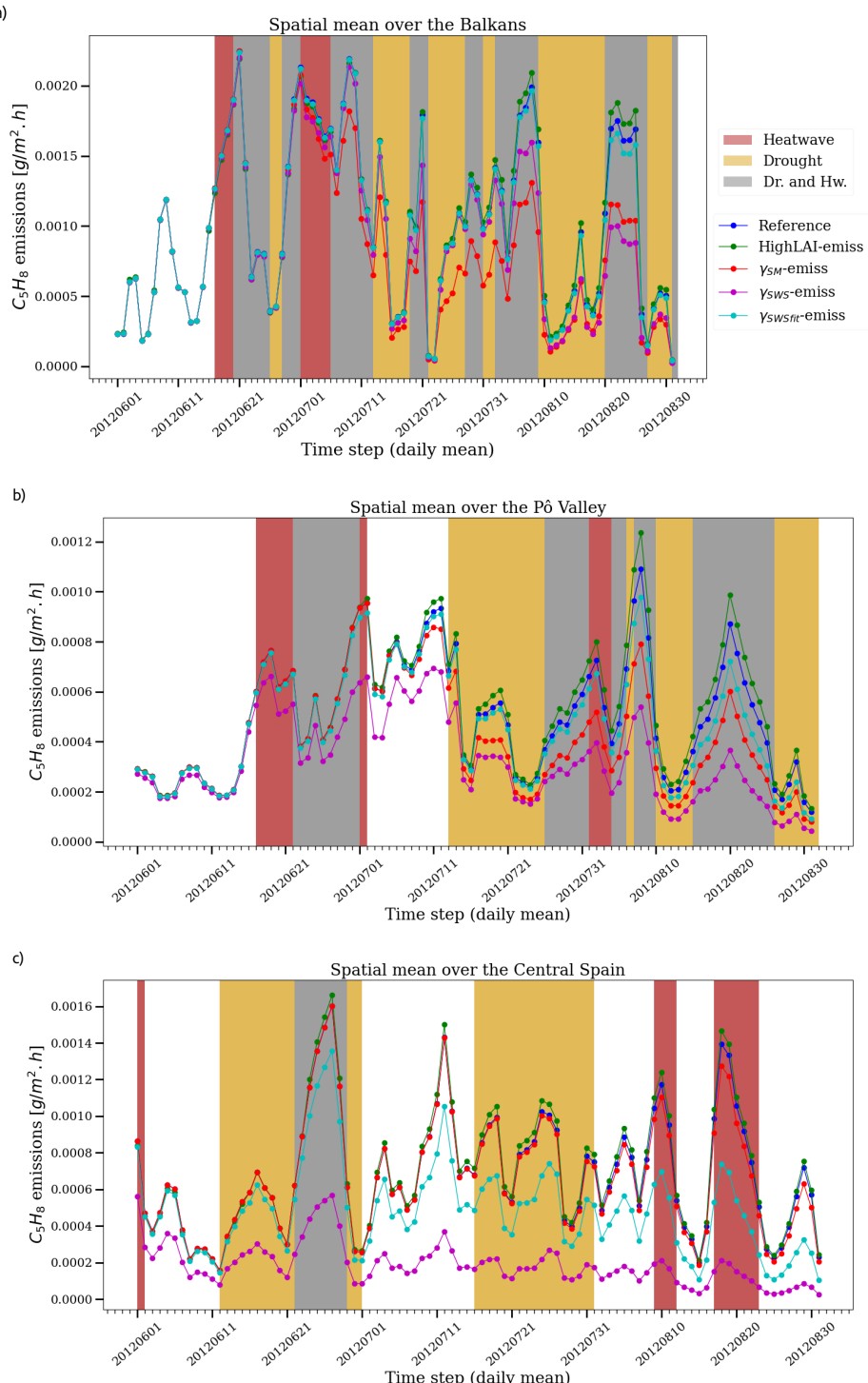

**Figure 3.** Daily mean $C_5H_8$ emissions $[g.m^{-2}.h^{-1}]$ during the summer 2012, spatially averaged over the Balkans (a), Pô Valley (b) and Central Spain (c) for the different MEGAN experiments. The colored bands highlight periods of droughts and heatwaves.

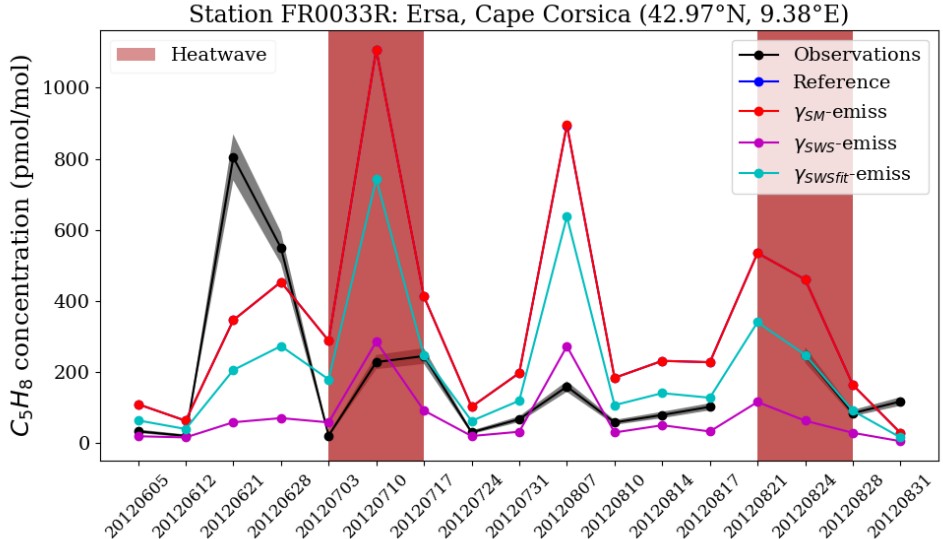

**Figure 4.** Observed surface $C_5H_8$ concentration [$pmol/mol$] during the summer 2012 at the Ersa station (FR0033R, Cape Corsica) from the EBAS data set, compared to the different simulated experiments undertaken by the MEGAN-CHIMERE model. The shaded curve in black represents the precision range of the measurements. The "Reference" and "$\gamma_{SM}$-emiss" experiments here have equal values. Maximum $PLA_{T2m}$ and $PLA_{SD}$ are +3.2°C and -0.02 of soil dryness index respectively.

explained by the LAI decrease prescribed from mid-July in the dry deposition scheme. In addition to that, the dry deposition velocity is characterized by a decreasing trend over summer imposed by the fixed phenology factor ($f_{phen}$).

Regarding the soil dryness, its effect is almost constant during summer (e.g. -12% over the Pô Valley). Based on measurements over Central Italy, Lin et al. (2020) computed a relative difference of about -50% between August 2004 (wet summer, ~0.8cm/s) and August 2003 (dry summer, ~0.4cm/s). However, summer 2003 was characterized by considerable heatwaves in

Italy (positive $PLA_{T2m}$) which might intensify the decrease. We emphasize that dry deposition velocity generally decreases during heatwaves (Fig. 6) as the near surface temperature is above the optimal temperature of stomatal conductance. Both effects related to droughts and (the most intense) heatwaves lead to a reduction in $O_3$ deposition. The sensitivity of dry deposition velocity to soil moisture can be considerably different from one deposition scheme to another. Using the same deposition scheme as in the present study, Anav et al. (2018) calculated an average decrease of 10% over Europe in dry $O_3$ deposition.

To the best of our knowledge, there is no study yet in the scientific literature that fully assess the $O_3$ dry deposition of CHIMERE against observations or its sensitivity to different meteorological forcings. Therefore, we have compared our results to the measured data available for summer 2012 at the Castelporziano station from the European Fluxex Database (Fig. 6). Based on the $PLA$ indicator, the Lazio region was affected by a severe drought all along the summer (mean $PLA_{SD}$ of +0.05). All simulated experiments overestimate $O_3$ deposition compared to observations. The "LAIdecr-dep/$f_{SWS}$-dep" experiment

has the smallest average bias (-0.19e-6 $g/cm^2$) and the highest correlation (R coefficient of 0.40). Such overestimation has also

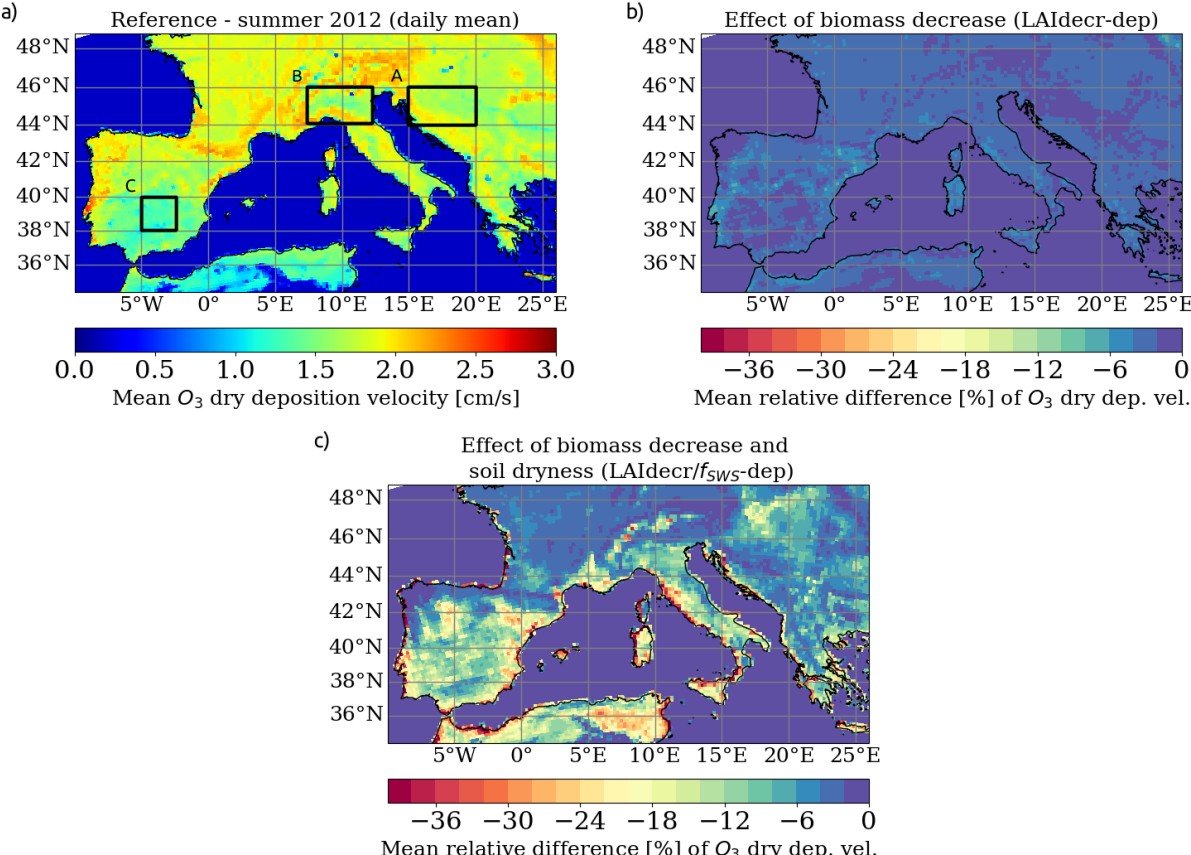

**Figure 5.** Summer mean (JJA) of daily mean $O_3$ dry deposition velocity [$cm/s$] for 2012 from the "Reference" simulation (a). Mean relative difference [%] of $O_3$ dry deposition velocity due to biomass decrease and soil dryness. The relative difference is computed between the "Reference" and "LAIdecr-dep" simulation to quantify the biomass decrease effect (b) and "LAIdecr-dep/$f_{SWS}$-dep" to quantify combined biomass decrease and soil dryness effect (c).

been calculated for models whose gas deposition scheme is based on Wesely (1989) "big leaf" parameterization (e.g. Michou et al., 2005; Huang et al., 2022). As the canopy conductance increases proportionally with the prescribed LAI (Emberson et al., 2000), this could be explained by an overestimation of the LAI that is almost two times larger than the mean LAI reconstructed from MODIS over this area. The importance of representing processes dynamically (as opposed to fixed parameters, especially for the non-stomatal conductance) is also highlighted in order to better simulate the diurnal deposition cycle, and so the daily
average values (Huang et al., 2022).

Finally, identifying soil dryness as a major driver of inter-annual variability of $O_3$ deposition velocity, Lin et al. (2019) emphasize that error in modelling the deposition may considerably rely on the ability of models to simulate accurately the precipitation distribution. They assessed the sensitivity of their deposition velocity scheme (Geophysical Fluid Dynamics Laboratory
LM3.0/LM4.0) to two different meteorological forcings and found a factor two over the Northern Europe.

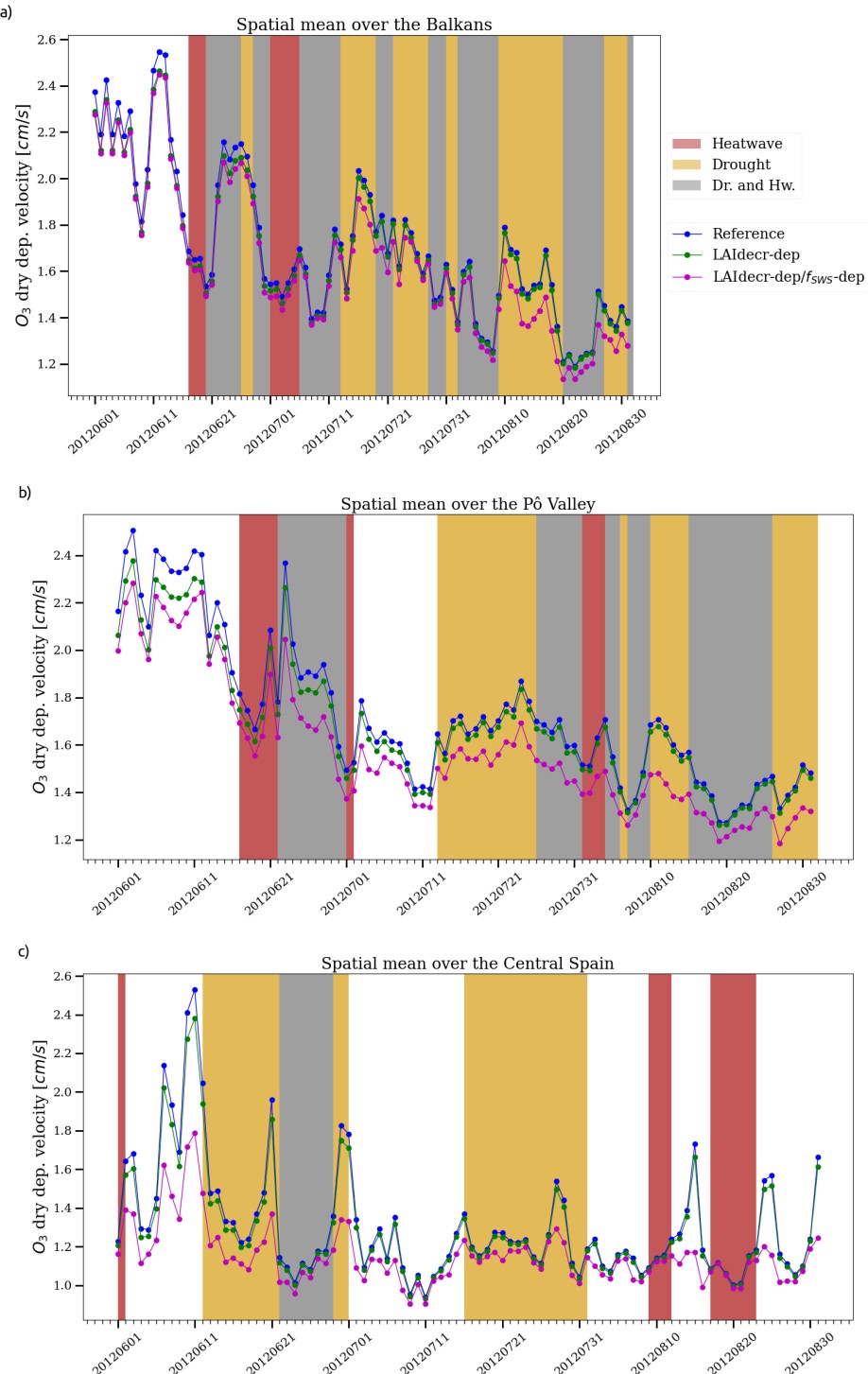

**Figure 6.** Daily mean $O_3$ dry deposition velocity [cm/s] during the summer 2012, spatially averaged over the Balkans (a), Pô Valley (b) and Central Spain (c) for the different CHIMERE experiments.

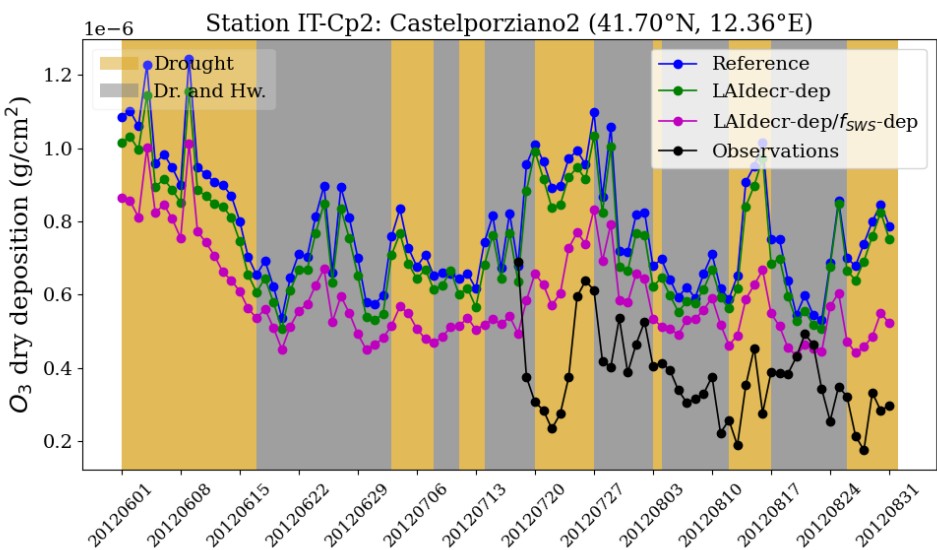

**Figure 7.** Observed $O_3$ deposition flux [g/cm$^2$] during the summer 2012 at the Castelporziano station (IT-Cp2, Lazio region) from the EFDC database, compared to the different simulated experiments undertaken by the CHIMERE model. Maximum $PLA_{T2m}$ and $PLA_{SD}$ are +3.88°C and +0.09 of soil dryness index respectively.

### 5.1.3 $O_3$ surface concentration

Figure 8a maps the mean surface $O_3$ concentration simulated by CHIMERE ("Reference" experiment) over the summer 2012. High concentrations (between 100 and 110$\mu g/m^3$) are located in the eastern part of the study area (e.g. the Balkans and Italy). The spatial distribution of mean surface $O_3$ shows similar patterns as the distribution of $PLA_{T2m}$ (SI: Fig. S2), highlighting
the critical role of the temperature. Surface $O_3$ remains high above the sea due to transport and the absence of deposition in the canopy (e.g. 120$\mu g/m^3$ over the Adriatic sea). The contribution of BVOC emissions on surface $O_3$ is also presented (Fig. 8b). BVOC emissions are known to be significant precursors of $O_3$ production in the Mediterranean region. It varies between 4% (e.g. Iberian Peninsula) and 22% (e.g. Northern Italy) in our study, that is included in the range of values reported in the scientific literature (e.g. Mertens et al., 2020). Finally, our areas of interest are located in rural regions, mostly characterized
by a low-$NO_x$ regime (SI: Fig. S7).

A simulation including drought effects on both $C_5H_8$ emissions and $O_3$ dry deposition has been conducted with MEGAN-CHIMERE ("$\gamma_{SWS}$-emiss & LAIdecr/$f_{SWS}$-dep" experiment). The resulting effect on surface $O_3$ is shown in Figure 8. On average during the summer (Fig. 8c), $O_3$ concentration is slightly higher over the continent (between +0.5% and +3.0%) due to the decrease of $O_3$ deposition as dominant effect, while it is slightly lower (between -1% and -1.5%) over the sea and ocean due
to the lower transport of $O_3$ precursors, compared to the "Reference" experiment. However, the $O_3$ increase over the Iberian Peninsula may be overestimated as the LAI reduction we applied to the whole domain (e.g. -20% for grass PFTs) is larger

compared to the variation of MODIS LAI in this specific region (SI: Fig. S3).

Drought effects on surface $O_3$ induced by $C_5H_8$ emission reduction is not constant. It is largest during combined heatwaves (i.e. when biogenic contribution is high). As a result, drought effects on $C_5H_8$ emissions can be dominant (compared to the effects on $O_3$ deposition) during simultaneous droughts and heatwaves, inducing a decrease of $O_3$ peaks by a few $\mu g/m^3$ both over continent and sea/ocean. The maximum absolute relative difference (Fig. 8d) reaches -5% over the Pô Valley and -14% along the Strait of Gibraltar. $O_3$ formation over the latter is favoured by large $NO_x$ shipping emissions.

Conducting a similar modelling experiment based on the $\gamma_{SM}$ from MEGANv3, Jiang et al. (2018) simulated a maximum absolute relative difference of surface $O_3$ of about -4% in August 2010. $O_3$ reduction (-10% on average) due to severe droughts was also measured in California over the period 2002-2015 (Demetillo et al., 2019). This was identified as being related to a steep decrease of $C_5H_8$ concentrations.

For each area of interest, the temporal evolution of surface $O_3$ based on the different CHIMERE experiments and the EEA observations (AQ e-Reporting) is presented (Fig. 9 for the Pô Valley and Fig. S8 for the Balkans and Central Spain in SI). Since the resulting effects of biomass decrease and soil dryness on surface $O_3$ are less than the bias between model and observations in our study (see Sect. 4), no simulation can be designated with certainty as the best fit.

## 5.2   Statistical variation during droughts and heatwaves

In this result section, a statistical analysis is performed using CHIMERE simulations and several observational data sets. The "all-emiss-dep" experiment has been chosen for simulating summers 2012-2014 because it includes drought and heatwave effects in the most comprehensive way. Moreover, the $C_5H_8$ emission approach "$\gamma_{SWSfit}$-emiss" has shown good performance compared to observations (Fig. 4), remaining rather conservative, not in the higher limit of the $C_5H_8$ reduction range.

Clusters of droughts and heatwaves (isolated or combined) are constructed based on the $PLA_{T2m}$ and $PLA_{SD}$ indicators, allowing to analyze the statistical variation of simulated $C_5H_8$ emissions, $O_3$ stomatal conductance and $O_3$ surface concentration for summers 2012-2014. We performed the same analysis on observations of $HCHO$ total column for summers 2005-2016 and $O_3$ for summers 2000-2016. Using those indicators, the following conditions were defined and grouped into clusters: "heatwaves or droughts", "heatwaves and droughts", "heatwaves and not droughts" and "droughts and not heatwaves". Normal conditions are defined as no drought and no heatwave.

### 5.2.1   $C_5H_8$ emissions

Figure 10 shows the distribution of $C_5H_8$ emission rates over the Southwestern Mediterranean for clusters of extreme weather events. On average, the daily maximum $C_5H_8$ emission is significantly higher (t-test, p < 0.01) during isolated and combined heatwaves than normal conditions (no heatwave nor drought) with a mean value of $0.16 \times 10^{-2}$ against $0.12 \times 10^{-2} g.m^{-2}.h^{-1}$. However, emissions during droughts have the same mean value as in normal conditions. During isolated droughts (cluster "drought and not heatwave"), the mean daily maximum $C_5H_8$ emission rate is lower than normal conditions ($-0.02 \times 10^{-2} g.m^{-2}.h^{-1}$, non-significant difference). These results are in general agreement with the observed $HCHO$ total column

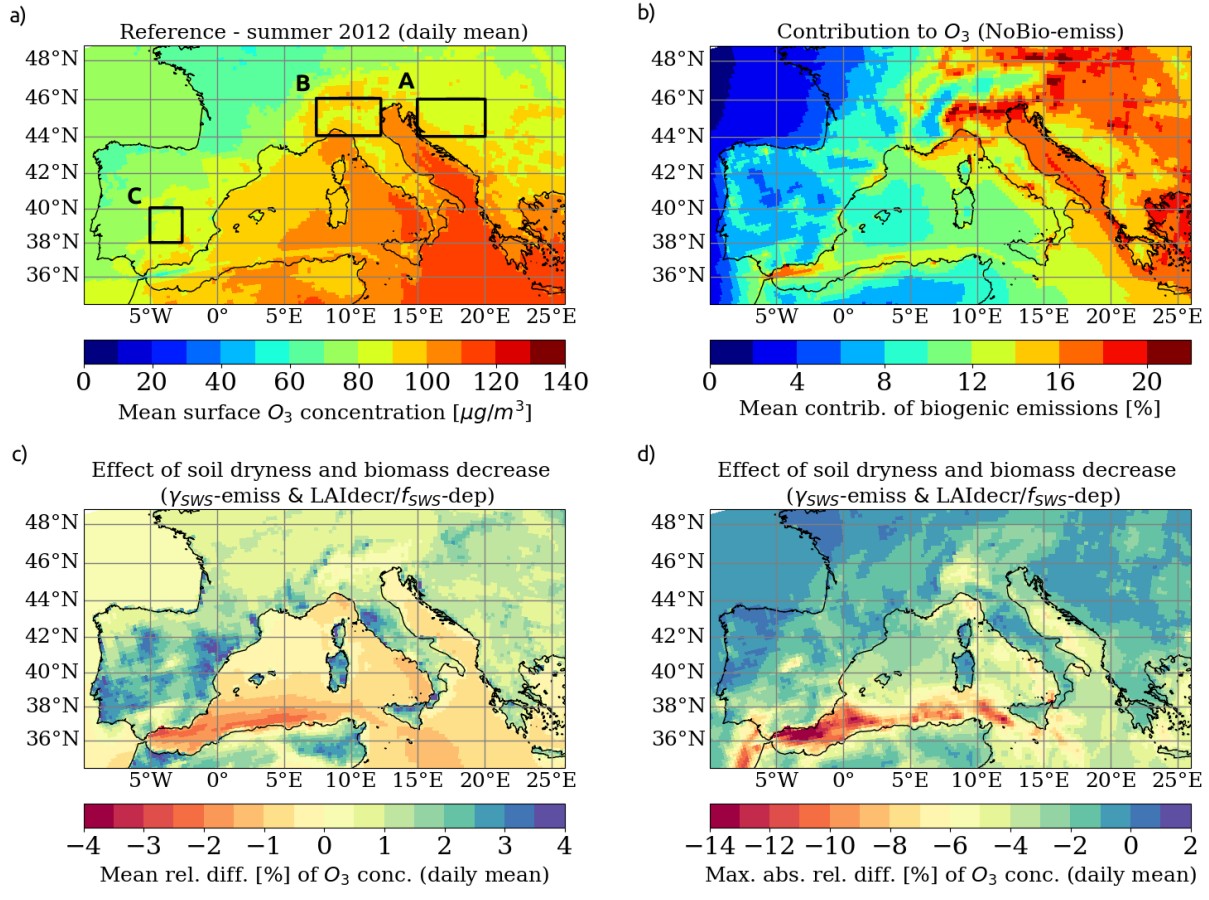

**Figure 8.** Summer mean (JJA) of daily mean $O_3$ surface concentration [$\mu g/m^3$] for 2012 from the "Reference" simulation (a). Mean contribution of biogenic emissions to $O_3$ surface concentration based on the "NoBio-emiss" experiment (b). Mean (c) and maximum absolute (d) relative difference [%] of $O_3$ surface concentration due to biomass decrease and soil dryness. The relative difference is computed between the "Reference" and "$\gamma_{SWS}$-emiss & LAIdecr-dep/$f_{SWS}$-dep" simulation from CHIMERE.

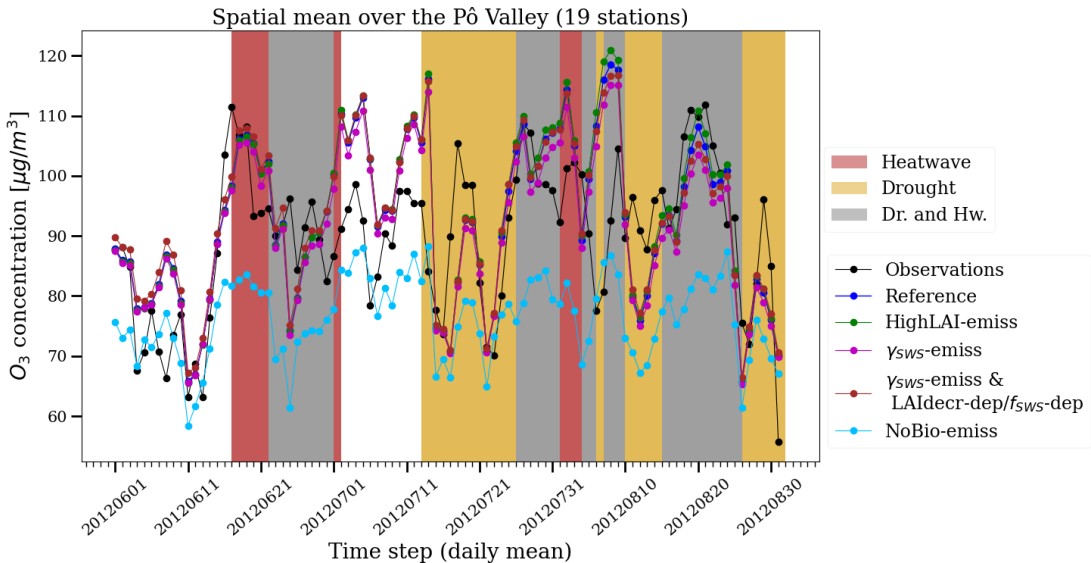

**Figure 9.** Daily mean $O_3$ surface concentration [$\mu g/m^3$] during summer 2012, spatially averaged over the Pô Valley from the EEA observations and the different CHIMERE experiments.

by satellite instrument, used as proxy of BVOC emissions variation (see Sect. 5.2.2).

Weather conditions are considerably different between droughts combined with heatwaves and those which are not (SI: Fig. S9). The combined (resp. isolated) droughts are characterized by a 2m temperature of 23.9°C (resp. 22.6°C), a shortwave radiation of $342.2W/m^2$ (resp. $296.0W/m^2$) and a cloud cover of 1.8% (resp. 3.1%). Those weather variables are used for the computation of the activity factors $\gamma_P$ and $\gamma_T$, thus directly affecting emissions. During isolated droughts, $\gamma_{LAI}$ (0.48) and $\gamma_{SWS}$ (0.76) are smaller than for normal conditions (0.51 and 0.89 respectively, significant difference for both). Nevertheless, this negative signal is not large enough for significant variation of the emission rates (compared to normal conditions).

Gathering worldwide data from experimental measurements of biogenic emissions under different climate drivers, the scientific review presented by Feng et al. (2019) (based on 74 articles) estimated to +53% the emission change of $C_5H_8$ during warm conditions and -15% during dry conditions. Those variations are close to what we simulated (+35% and -13% respectively).

### 5.2.2 $HCHO$ total column

$HCHO$ is a product of oxdation of VOCs. Based on the "Reference" and "NoBio-emiss" CHIMERE simulations, we computed that biogenic emissions contribute between 60 and 80% of the $HCHO$ concentration over Southwestern Europe. Variations of $HCHO$ concentration may therefore be used as indicator of BVOC emission variations during droughts and heatwaves. This allows us to use satellite observations of $HCHO$, which is particularly interesting due to the lack of in-situ data. Observations of the $HCHO$ total column from the OMI instrument are used.

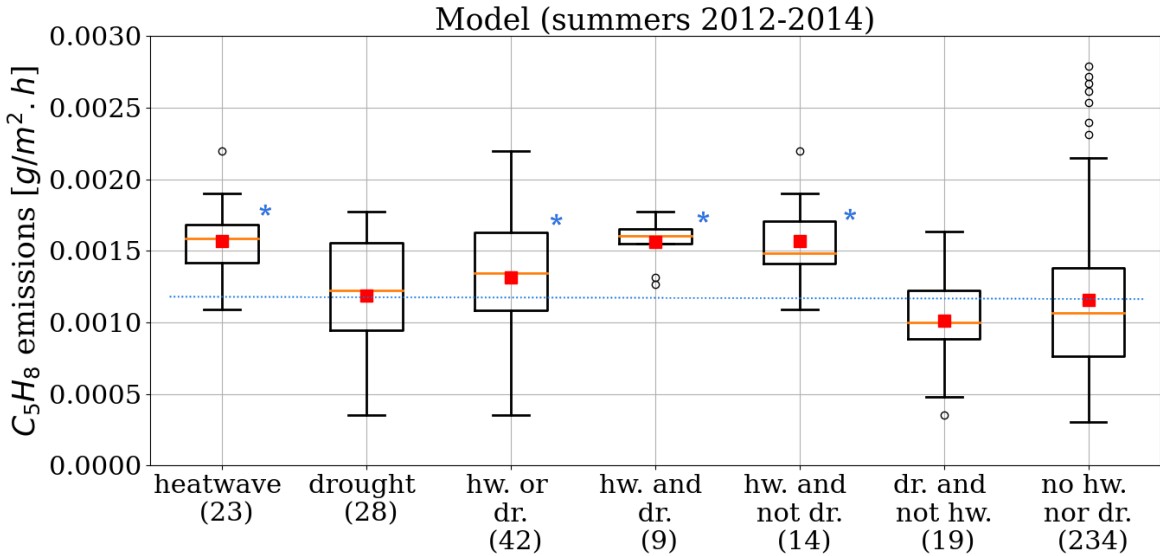

**Figure 10.** Daily maximum $C_5H_8$ emission rate $[g.m^{-2}.h^{-1}]$ simulated by the MEGAN model ("$\gamma_{SWSfit}$-emiss" experiment) over South-western Europe, clustered by identified extreme weather events ($PLA_{T2m}$ and $PLA_{SD}$ indicators from RegIPSL). The number of days is indicated in parentheses. The analyzed period is June-July-August 2012, 2013 and 2014, covering a total of 276 days. The red squares show the mean of the distribution and the black circles are the outliers. The blue dotted line indicates the mean value of the normal conditions ("no hw. nor dr." cluster) and the blue stars if the mean value of the considered cluster is significantly different (t-test, at least p < 0.1) from the normal conditions. The box covers the InterQuartile Range (IQR) between Q1 (25th percentile) and Q3 (75th percentile). The lower whisker is limited to a statistical minimum (Q1-1.5*IQR) and the upper one to a statistical maximum (Q3+1.5*IQR).

Table 3 presents the average difference of $HCHO$ ($\Delta HCHO$) during extreme events compared to normal conditions (no heatwave nor drought), based on the OMHCHOd product. Over summers 2005 to 2016, $HCHO$ is significantly higher during heatwaves for the Balkans, Pô Valley and Central Spain (+15% on average). $\Delta$HCHO is also positive during droughts but to a lesser extent (+3% on average, non-significant for Central Spain only). However, isolated droughts induce a significant

decrease for the Balkans (-7%), Pô Valley (-6%) and Central Spain (-6%). Those results are consistent with the variation of $C_5H_8$ emissions simulated by MEGAN when a soil moisture parameter is considered (see Sect. 5.2.1). Wang et al. (2021a) also report a significant $HCHO$ column decrease (up to -30%) induced by a prolonged drought in a forested area in China. The observed and simulated $HCHO$ total columns have been compared for the "$\gamma_{SWSfit}$-emiss" experiment over summer 2012. Time series showing $HCHO$ evolution for the three areas of interest are shown in SI (Fig. S11). Simulated $HCHO$

columns are generally higher than those from observations, especially for the Balkans. As mentioned in Sect. 2.2, uncertainty on the observations is large (between 30 and 100%) with large spatial variability (about a factor of two larger than in CHIMERE). HCHO products from OMI also present high uncertainty and in particular a systematic low mean bias (20-51%) (Zhu et al., 2016).

$HCHO$ differences could be due to wrong specifications of land cover, and thus of the EF of $C_5H_8$ (e.g. Curci et al., 2010).
Temperate tree PFTs are characterized by high $C_5H_8$ emission rates (respectively 10 000 and $600\mu g.m^{-2}.h^{-1}$ for broadleaf and needleleaf types), compared to grassland ($800\mu g.m^{-2}.h^{-1}$) or cropland ($1\mu g.m^{-2}.h^{-1}$) (Guenther et al., 2012). After aggregation of the USGS land cover classes, the vegetation type assumed in CHIMERE in the Balkans, for instance, is 57% of forest cover, 9% grassland and 33% cropland. Using the MODIS MCD12 product (Friedl et al., 2010), we find a different distribution, with 30% of forest cover, 25% grassland and 31% cropland (SI: Fig. S10). The choice as well as the temporal evolution of the land cover database are crucial for the calculation of $C_5H_8$ emissions (Chen et al., 2018). Despite the use of satellite data for land cover, significant uncertainties remain in the calculation of $C_5H_8$ emissions due to the classification of vegetation types and species (Opacka et al., 2021).

The comparison of $HCHO$ total column variations for CHIMERE and OMI during droughts and heatwaves relies on a limited number of cases. It is therefore difficult to support conclusions with a sufficient level of certainty. Nevertheless, results suggest that CHIMERE simulations are more sensitive to temperature than the OMHCHOd OMI observations (+52% and +28% during heatwaves respectively, averaged over the three areas of interests). Since the summer 2012 was affected by long agricultural droughts, the inclusion of a soil dryness parameter in CHIMERE ("$\gamma_{SWSfit}$-emiss" experiment) reduces the simulated $HCHO$ peaks (SI: Fig. S11). However, the mean simulated $\Delta HCHO$ during heatwaves and droughts is similar in the "$\gamma_{SWSfit}$-emiss" and "Reference" experiments (decrease of -2% and -1% respectively). The temporal correlation does not vary significantly either (e.g. R coefficient around 0.5 for the Balkans).

Finally, the observed variations of total $HCHO$ over summers 2005-2006 and the three areas considered here were also computed with OMI-BIRA retrieval. During heatwaves, both show a significant increase: +31% for OMI-BIRA (and +15% for OMHCHOd). The increase is lower during droughts: +13% (and +3%). The variation becomes slightly negative during isolated droughts: -2% (and -6%). This comparison highlights the uncertainty in the satellite observations but also that the general behavior is consistent in both retrievals and similar to what was obtained for 2012 using CHIMERE.

### 5.2.3  $O_3$ stomatal conductance

Surface weather conditions are critical for the stomatal conductance and therefore influence the dry deposition velocity. Figure 11 shows the maximum daily stomatal conductance of $O_3$ ("LAIdecr/$f_{SWS}$-dep" CHIMERE experiment) clustered by simulated extreme weather events and averaged over the Western Mediterranean. The same analysis has been performed on the dry deposition velocity and signals induced by extreme weather events are similar.

Droughts and heatwaves (isolated or combined) induce a significant decrease of the $O_3$ stomatal conductance, quantified at -25% for heatwaves and -35% for droughts compared to normal conditions. The activity factors mainly affected by droughts and heatwaves are $f_{temp}$, $f_{VPD}$ and $f_{SWS}$.

The variation of $f_{temp}$ during heatwaves depends on the magnitude of the events and so on their location, since the percentile of $PLA$ indicator is defined for each grid cell. Over the Pô Valley and Balkans for instance, most heatwaves are characterized by temperatures close to optimal values of stomatal conductance that is fixed around 24°C (SI: Fig. S6). However, for those occurring in Central Spain (between 30° and 32°C), $f_{temp}$ decreases by 7% compared to normal conditions. The temperature

**Table 3.** Variation of HCHO total atmospheric column [$molecules/cm^2$] ($\Delta$HCHO) due to heatwaves, droughts and isolated droughts in comparison to normal conditions (no heatwaves nor droughts) for summers (JJA) between 2005-2016 (measurements at 1pm). Summer 2012 is compared with CHIMERE simulations. Results are computed for each pixels and averaged over areas of interests: the Balkans, Central Italy and Central Spain. Stars mean that the difference with normal conditions is statistically significant (t-test, p < 0.1).

| | OMI (2005-16) | OMI (2012) | CHIMERE (2012) | |
| --- | --- | --- | --- | --- |
| | | | Reference | $\gamma_{SWSfit}$-emiss |
| Balkans (norm. cdt.) | $5.7 \times 10^{14}$ $mol./cm^2$ | $4.0 \times 10^{14}$ $mol./cm^2$ | $7.6 \times 10^{14}$ $mol./cm^2$ | $7.6 \times 10^{14}$ $mol./cm^2$ |
| $\Delta$ with heatwaves | + 17 %* | + 64 %* | + 94 %* | + 92 %* |
| $\Delta$ with droughts | + 6 %* | + 54 %* | + 63 %* | + 62 %* |
| $\Delta$ with isolated droughts | - 7 %* | + 32 %* | + 10 %* | + 9 %* |
| | | | | |
| Pô Valley (norm. cdt.) | $5.9 \times 10^{14}$ $mol./cm^2$ | $5.3 \times 10^{14}$ $mol./cm^2$ | $7.6 \times 10^{14}$ $mol./cm^2$ | $7.5 \times 10^{14}$ $mol./cm^2$ |
| $\Delta$ with heatwaves | + 16 %* | + 17 %* | + 36 %* | + 34 %* |
| $\Delta$ with droughts | + 2 %* | + 11 %* | + 24 %* | + 23 %* |
| $\Delta$ with isolated droughts | - 6 %* | - 1 % | - 3 %* | - 3 %* |
| | | | | |
| Central Spain (norm. cdt.) | $5.2 \times 10^{14}$ $mol./cm^2$ | $4.8 \times 10^{14}$ $mol./cm^2$ | $6.1 \times 10^{14}$ $mol./cm^2$ | $5.8 \times 10^{14}$ $mol./cm^2$ |
| $\Delta$ with heatwaves | + 12 %* | + 2 % | + 28 %* | + 25 %* |
| $\Delta$ with droughts | + 0 % | - 7 %* | + 9 %* | + 8 %* |
| $\Delta$ with isolated droughts | - 6 %* | - 11 %* | - 3 %* | - 3 %* |

limit before complete stomatal closure is set at 40°C. Therefore exceptional heatwaves occurring in Southern Spain for instance could lead quickly to an accumulation of $O_3$ at the surface. $f_{VPD}$ that depends both on temperature and relative humidity; significantly decreases during droughts and heatwaves (e.g. -6% averaged over the Southwestern Europe). Finally, $f_{SWS}$ is the factor dominating the signal of variation of stomatal conductance. At the Southwestern Europe scale, this factor is the lowest during isolated droughts with a mean decrease of -35%.

### 5.2.4 $O_3$ surface concentration

Figure 12 shows the distribution of the observed (summers 2000-2016 and 2012-2014) and simulated (summers 2012-2014) daily maximum surface $O_3$ concentrations over Southwestern Europe for each cluster of extreme events. Observed $O_3$ (Fig. 12a) is significantly (t-test, p < 0.01) higher during heatwaves (+18$\mu g/m^3$) and droughts (+9$\mu g/m^3$) than during normal conditions. Considering all droughts over the United States of America, Wang et al. (2017) also computed a mean increase in surface $O_3$ (+17% compared to the average). During isolated droughts in our study area, daily maximum $O_3$ is larger (+4$\mu g/m^3$) but the difference is non-significant.

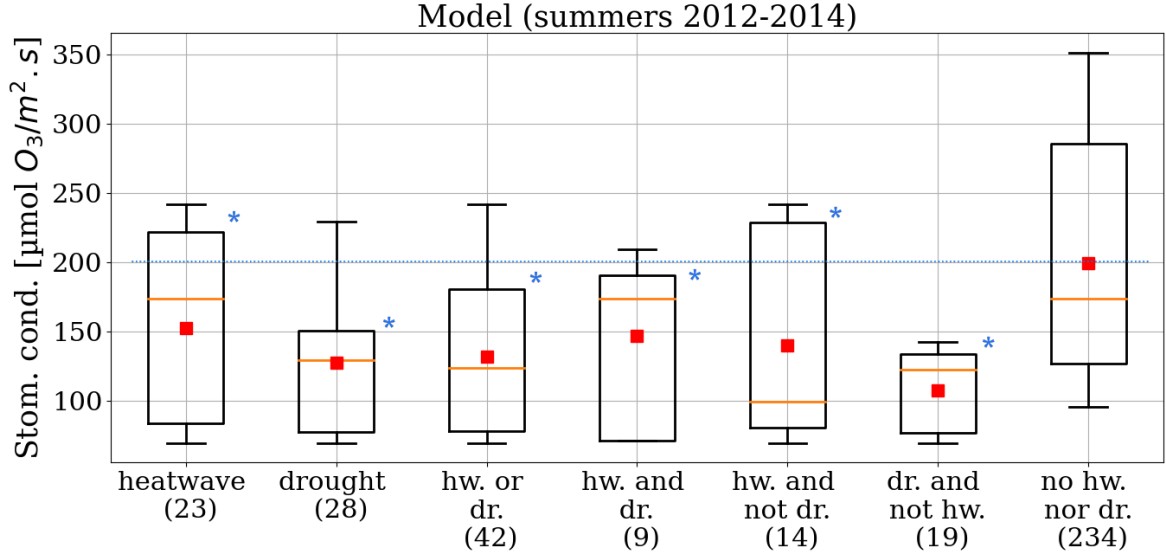

**Figure 11.** Same as Figure 10 with the simulated $O_3$ stomatal conductance [$\mu mol O_3.m^{-2}.s^{-1}$] by the CHIMERE model ("LAIdecr/$f_{SWS}$-dep" experiment).

The distribution of the simulated surface $O_3$ ("all-emiss-dep" experiment) by extreme events over the period 2012-2014 (Fig. 12c) presents similar signals but of lower magnitude: $+9\mu g/m^3$ during heatwaves, $+3\mu g/m^3$ during droughts and non-significant difference during isolated droughts compared to normal conditions. Based on the results discussed above, the difference between the "heatwave" and "isolated drought" cluster could be explained by the different conditions of biogenic emissions, dry deposition, temperature and light. However, observations over the same period (Fig. 12b) present a significant increase of the daily maximum $O_3$ during isolated droughts ($+9\mu g/m^3$), unlike what we simulated. The $C_5H_8$ emission reduction effect during such extreme event could be counterbalanced in a larger extent by the $O_3$ dry deposition decrease. It could also be explained by an underestimated impact of the enhanced photochemistry in the simulations, as we simulated favorable weather conditions during both combined and isolated droughts.

In summary, the variation of canopy-troposphere interactions (simulated by the MEGAN and CHIMERE models) during droughts and heatwaves is characterized by a consistent signal with respect to $O_3$ observations (except for the isolated droughts over summers 2012-2014). Meteorological conditions are critical for the $O_3$ budget especially during summer droughts and heatwaves. In addition to uncertainties in the modelling of precursor emissions and $O_3$ deposition (as mentioned above), differences between observations and simulations may also rely on meteorological uncertainties, such as the diurnal temperature cycle (see Sect. 4) and the Planetary Boundary Layer Height (PBLH). The night-time representation of the PBLH is often misrepresented in the WRF model (e.g. Chu et al., 2019).

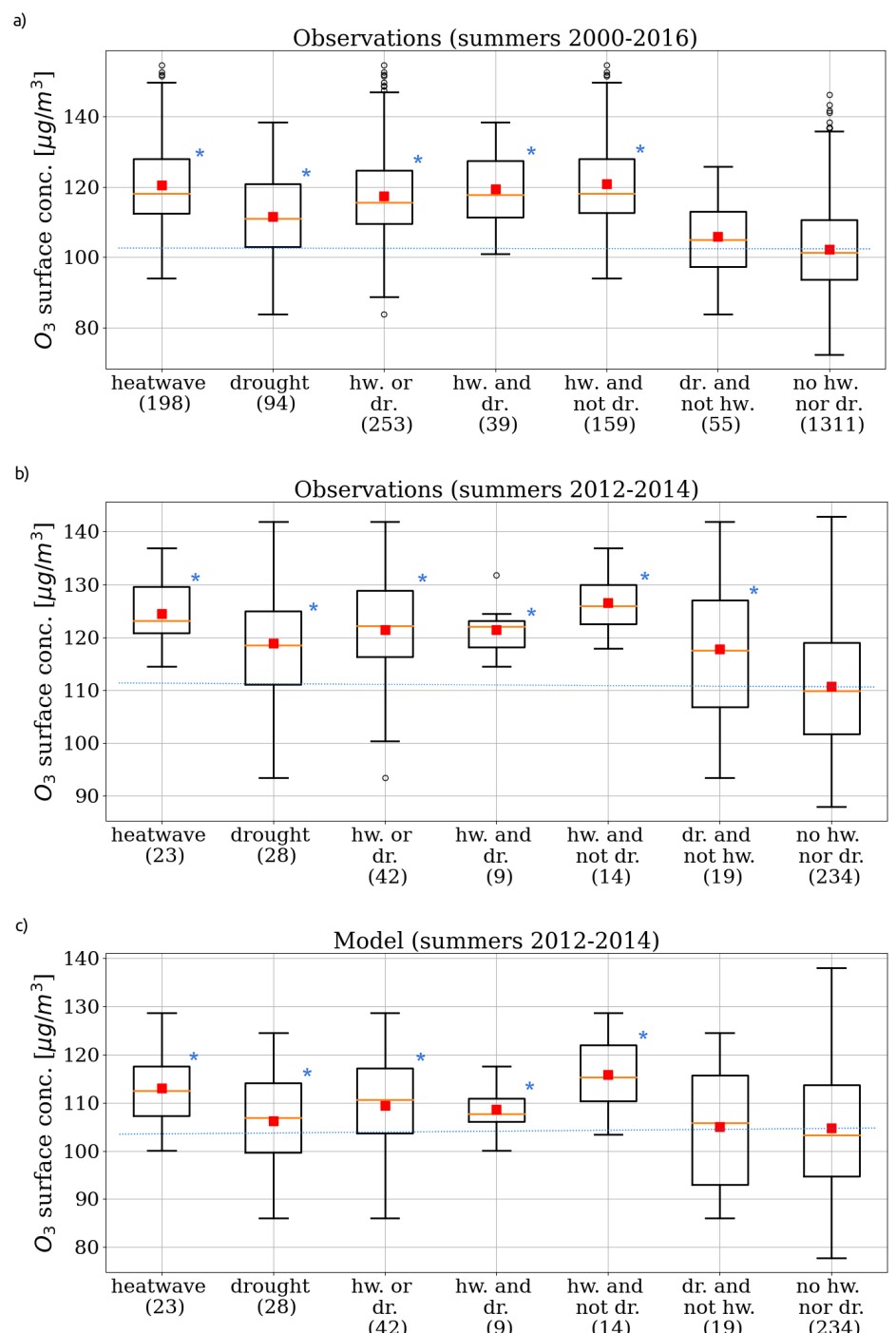

**Figure 12.** Same as Figure 10 with the observed surface $O_3$ concentration $[\mu g/m^3]$ over the summers 2000-2016 (a) and 2012-2014 (b), and with the simulated surface $O_3$ ("all-emiss-dep" experiment) over the summers 2012-2014 (c).

## 5.3 Threshold level exceedance of $O_3$

In the European Union, the air quality standard for $O_3$ exposure is set at a daily maximum concentration of $120 \mu g/m^3$ (8h-average) (EEA, 2020). Figure 13 shows the number of days above this threshold based on the AQ e-Reporting observations and the CHIMERE "all-emiss-dep" simulations for summers 2012, 2013 and 2014. Summers 2012 and 2013 present a large number of days exceeding the standard concentration. Even if exceedances occur in similar regions for observations and simulations, the number of days is generally larger in the observations. This is due to the overall underestimated daily maximum in CHIMERE compared to observations (see Sect. 4). For example in the Pô Valley, that is the most affected region in Southwestern Europe, around 60 exceeding days were observed and 50 were simulated over the summer 2012. This region is known for its highly polluted air ($O_3$ and its precursors) due to high anthropogenic emissions and unfavorable topographic and meteorological conditions for pollutants dispersion (e.g. Bigi et al., 2012). Other regions affected by $O_3$ peaks can be highlighted: Southeastern France, central Spain and central Italy for both summers 2012 and 2013.

Table 4 presents the distribution characteristics of stations with at least one day above the EU standard during summer period. On average over 2000-2016, that concerns 54% of EEA stations over the Western Mediterranean. The average number of exceedance days per station is 27 (almost a third of the summer period). Summer 2012 is above the 2000-2016 average, with 61% of stations affected and 28 days on average per station. For the same year, co-located values from CHIMERE simulations ("all-emiss-dep" experiment) are lower: 57% of stations and 18 days on average.

Over all summers between 2000 and 2016, 34% of the exceeding days occurred during heatwaves (with a mean exceeding value of $24 \mu g/m^3$) and 27% during droughts ($+18 \mu g/m^3$). The number of days decreases by 13% if we consider only the isolated droughts (14%, $+15 \mu g/m^3$). Summer 2012 was affected by exceptional droughts and heatwaves, resulting in high $O_3$ pollution. Around 80% of days above the EU threshold occurred during heatwaves or droughts for both observations and simulations.

## 6 Discussion and conclusions

The analyses presented in this study were organized around two main objectives. The first one was to assess the sensitivity of biogenic emissions, $O_3$ dry deposition and surface $O_3$ to the biomass decrease and soil dryness effect in a CTM model. The extremely dry summer 2012 was chosen and simulations were performed using the MEGAN v2.1 and CHIMERE v2020r1 model. We showed that the soil dryness parameter is critical during drought events, decreasing considerably the $C_5H_8$ emissions and $O_3$ dry deposition velocity. This effect has a larger impact than the biomass decrease. However, the resulting effect on surface $O_3$ remains limited.

In addition to the soil moisture activity factor used in MEGAN v2.1 that is mainly based on the wilting point ($\gamma_{SM}$), we proposed an innovative activity factor based on a soil water stress function ($\gamma_{SWS}$) simulated by the land surface and vegetation model ORCHIDEE. The latter induces a larger reduction of $C_5H_8$ emissions with more homogeneous patterns that follow the drought indicator. Furthermore, we adjusted this factor with a function fitted ($\gamma_{SWSfit}$) from experimental measurements in Bonn et al. (2019). By comparing the simulated surface concentration of $C_5H_8$ with observations at the Ersa measurement site

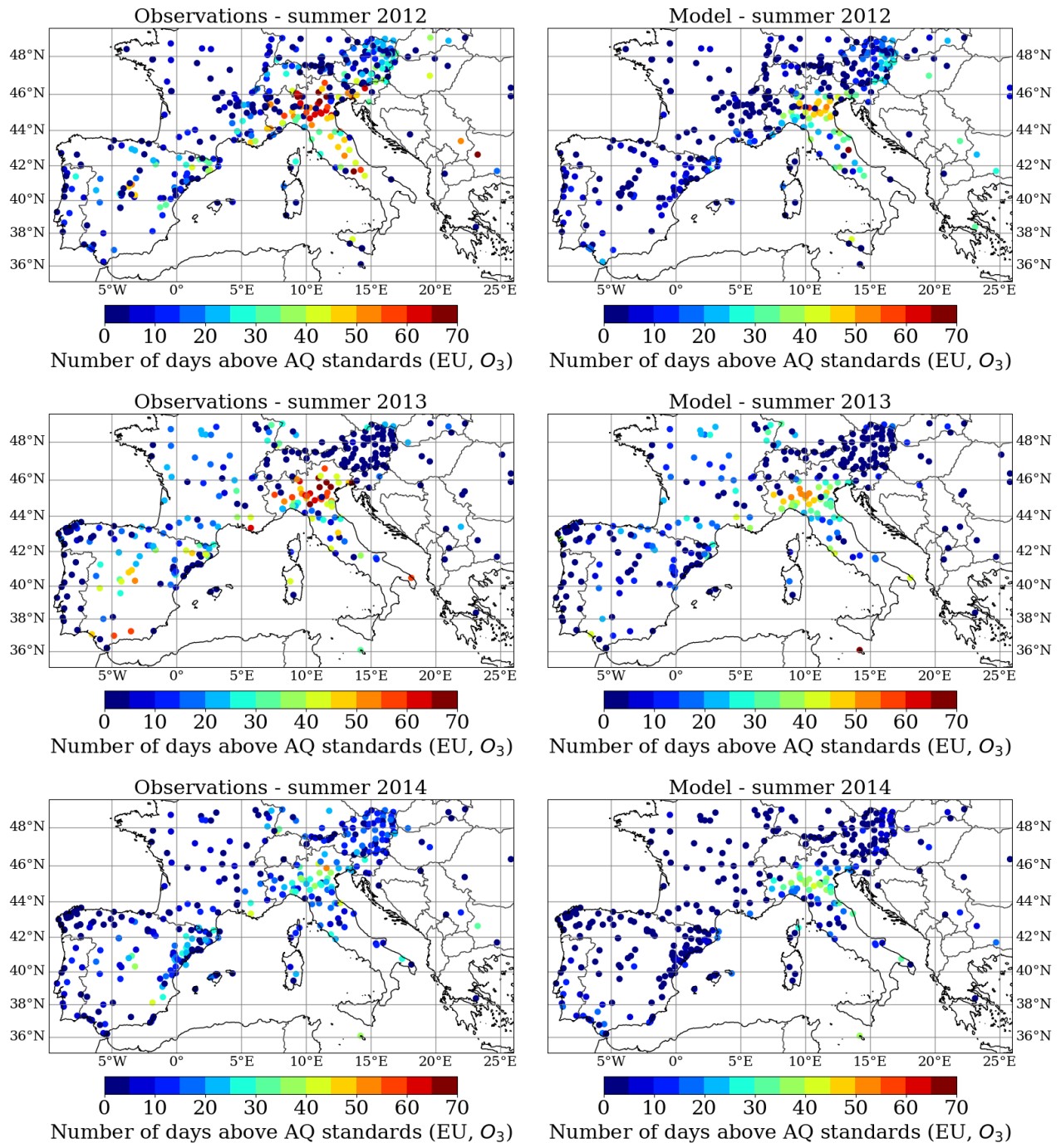

**Figure 13.** Number of days above the air quality threshold value from the European Union for surface $O_3$ (maximum daily 8 hour mean of 120 $\mu g/m^3$) over the summer 2012, 2013 and 2014 (JJA, 92 days in total). EEA observations (left column) are compared to CHIMERE ("all-emiss-dep") simulations (right column).

**Table 4.** Average percentage of stations with at least one exceedance day (first row) regarding EU standard during summer (JJA). Considering only those stations, the second row shows the average number of exceeding days per station and the lower rows the average distribution of days above the EU standard in function of extreme weather events. The mean exceedance concentration is indicated in parenthesis.

| | EEA (2000-2016) | EEA (2012) | CHIMERE (2012) |
|---|---|---|---|
| Average fraction of stations (over all): | 54% | 61% | 57% |
| Average number of exc. days / station: | 27 days | 28 days | 18 days |
| | | | |
| Average distribution of exc. days: | | | |
| Normal conditions | 52% ($+17\mu g/m^3$) | 20% ($+15\mu g/m^3$) | 21% ($+15\mu g/m^3$) |
| Heatwaves or droughts | 48% ($+22\mu g/m^3$) | 80% ($+18\mu g/m^3$) | 79% ($+17\mu g/m^3$) |
| Heatwaves | 34% ($+24\mu g/m^3$) | 58% ($+20\mu g/m^3$) | 65% ($+18\mu g/m^3$) |
| Isolated heatwaves | 21% ($+24\mu g/m^3$) | 16% ($+21\mu g/m^3$) | 18% ($+16\mu g/m^3$) |
| Droughts | 27% ($+18\mu g/m^3$) | 64% ($+18\mu g/m^3$) | 61% ($+17\mu g/m^3$) |
| Isolated droughts | 14% ($+15\mu g/m^3$) | 22% ($+14\mu g/m^3$) | 14% ($+15\mu g/m^3$) |

(Corsica), $\gamma_{SWSfit}$ showed promising results. However, such evaluation should be carried out over several sites and several years in order to determine with larger certainty the added value of this approach.

The second objective of this study was to quantify the variation of surface $O_3$ over the Southwestern Europe during agricultural droughts, combined or not with heatwaves. Those extreme weather events were identified based on the RegIPSL model using the PLA indicator. During the 2000-2016 period, 59% of summer drought days were not accompanied by heatwaves (isolated droughts). For the summers 2012-2014, analyzed more specifically in this study, the 2m temperature is on average 5.5% lower during isolated droughts compared to all droughts, the shortwave radiation is 13.5% lower and the cloud fraction is 42% higher. As a result, surface $O_3$, but also BVOC emissions and $O_3$ dry deposition velocity, are substantially different if the drought considered is accompanied by a heatwave or not.

Based on a cluster approach using the PLA indicator, we showed that observed surface $O_3$ (summers 2000-2016) is larger by $+18\mu g/m^3$ in daily maximum during heatwaves and by $+9\mu g/m^3$ during droughts, compared to normal conditions. Despite a difference of several $\mu g/m^3$, CHIMERE correctly simulates the variations of $O_3$ concentration between the clusters of extreme events. The overall increase of surface $O_3$ during both heatwaves and droughts would be explained by $O_3$ precursor emission enhancement, $O_3$ dry deposition decrease and favorable weather conditions, so that all these mechanisms lead to an accumulation of $O_3$. However, we simulated a decrease of $C_5H_8$ emissions during isolated droughts, resulting in a non-significant difference of surface $O_3$ compared to normal conditions (from both observations and simulations). Despite a significant bias between the $HCHO$ total columns simulated by CHIMERE and observed by OMI, the satellite data confirm an average increase of $HCHO$ (+3% to +13% depending of the product considered) over our three regions of interest for all droughts and

a decrease (-2% to -6%) for isolated droughts.

Finally, almost half of summer days (2000-2016 period) exceeding the EU standard of $O_3$ for air quality in Southwestern Europe occurred during droughts or heatwaves. However, this percentage can increase (up to 80%) for exceptionally dry and hot summers, like in 2012. Only 14% of the exceedance days occurred during isolated droughts.

The implementation of dynamical effects of droughts in the MEGAN - CHIMERE model contributes to a better representation of biosphere-atmosphere interactions. However, comparisons between simulated and observed surface $O_3$ still show large discrepancies. Important uncertainties appear to be related to BVOC emissions (especially due to the land cover classification), to $NO_x$ concentrations for which CHIMERE presents limited performance scores of validation, to $O_3$ deposition and finally to meteorological conditions (e.g. temperature and PBLH). For instance, the simulated daily maximum temperature that is underestimated in the Northern Mediterranean compared to observations (see Sect. 4), may induced a decrease of $O_3$ precursors, especially as the emission–temperature relationship of BVOCs is exponential, with an optimal temperature for $C_5H_8$ species (Guenther et al., 1993). Such uncertainties need to be addressed to improve the simulation of $O_3$ during the summer, and especially over the Southwestern Europe.

In addition, the critical role of soil $NO$ emissions in $O_3$ production is increasingly studied (e.g. Romer et al., 2018), especially in rural areas (e.g. Sha et al., 2021). As hydro-climatic conditions are critical for soil $NO$ emissions, the $O_3$ budget during droughts and heatwaves is likely to be significantly influenced by soil $NO$ emissions. In this study, dependence to soil dryness for $NO$ emissions is not included. Emission pulses can occur when rain follows a drought and emission factors are higher with dry soil than with wet soil (Steinkamp and Lawrence, 2011; Weng et al., 2020).

Some recent knowledge on fundamental processes that allow a better representation of surface-atmosphere interactions during extreme weather events is not yet integrated by much, if not all, of the modelling community. Among the many examples are the increased emission of monoterpenes and sesquiterpenes during the development of drought (e.g. Bonn et al., 2019; Peron et al., 2021) or the in-canopy chemistry that is ignored or approximated by the "big leaf" model approach (e.g. Clifton et al., 2020a). Several studies show that non-stomatal conductance in the Mediterranean counts as much as stomatal conductance in the $O_3$ sink budget (e.g. Gerosa et al., 2009; Sun et al., 2022). Moreover, non-stomatal conductance also seems to be significantly affected during droughts and heatwaves. In contrast to stomatal conductance, droughts and heatwaves should have opposite effects: an increase in non-stomatal conductance during heatwaves and a decrease during droughts (Wong et al., 2022). However, such changes are not taken into account in the EMEP deposition scheme implemented in CHIMERE. There is a real need for a better representation of stomatal conductance in deposition schemes.

In conclusion, we provide in this paper a detailed analysis of the drought and heatwave effects on modeled biosphere-troposphere interactions controlling surface $O_3$ concentration, supported by several observational data sets. Heatwaves, and droughts in a lower extent, induce a significant increase of the surface $O_3$. Soil dryness and biomass decrease, as specific effects of droughts, are critical for the variation of $C_5H_8$ emissions and $O_3$ dry deposition over the Southwestern Europe. We emphasize the need for a more dynamical representation of interactions between vegetation, meteorology and atmospheric chemistry in models in order to improve the simulation of summer $O_3$.

*Acknowledgements.* This work was granted access to the HPC resources of TGCC under allocation 10274 made by GENCI (Grand Equipement National de Calcul Intensif) and funded by Sorbonne Université (SU) and the Centre National d'Etudes Spatiales (CNES). We acknowledge the EEA to provide the AQ e-Reporting data set and Guillaume Siour for the data extraction and preparation. The MCD153AH and OMH-CHOd products were retrieved from the NASA EOSDIS Land Processes Distributed Active Archive Center (LPDAAC) and USGS Earth Resources Observation and Science (EROS) Center. Finally, we acknowledge the NCEP for the reanalysis meteorological data and the Norwegian Institute for Air Research (NILU) for the EBAS database.

*Data availability.* The data set of indicators of heatwaves and agricultural droughts ($PLA_{T2m/SD}$) is freely available at https://thredds-x. ipsl.fr/thredds/catalog/HyMeX/medcordex/data/Droughts_Heatwaves_1979_2016/catalog.html. The data from AQ e-Reporting can be found at https://www.eea.europa.eu/data-and-maps/data/aqereporting-9/aq-ereporting-products, from MODIS and OMI instrument at https://lpdaac. usgs.gov/data/, from NCEP at http://www.ncep.noaa.gov, from EBAS at https://ebas.nilu.no/data-access/ and from EFDC at http://www. europe-fluxdata.eu/.

*Competing interests.* The authors declare no competing financial interest.

*Author contributions.* AG, ST and JP: Conceptualization. AG, ST, AC, JP, AE and JL: Methodology. AG: Visualization. AG and ST: Writing of the original draft. AG, ST, AC, JP, AE and JL: Review and editing. All authors have read and agreed to the final version of the manuscript.

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
