# Peer review of "Biogenic isoprene emissions, dry deposition velocity and surface ozone concentration during summer droughts, heatwaves and normal conditions in Southwestern Europe"

_EGUsphere, 2022_

## Referee Comment (RC1)

This manuscript studies the biospheric impacts of heatwave and drought on surface ozone over southwestern Europe, using well-established numerical models and methodologies. As chemical transport models cannot fully resolve the biosphere, omitting some relevant effects and processes is acceptable, provided that their potential impacts on the results and implications are well-discussed. This part is somewhat missing in this manuscript, and should be addressed.

Major issues:

Soil NO emission: Increasing evidence shows that the temperature dependence of soil NO emission is an important part of $O_3$ production, especially in rural and agricultural regions (Oikawa et al., 2015; Romer et al., 2018; Sha et al., 2021), which cover a significant portion of the studies domain. Since this study focus on heatwaves and droughts, which directly affect two of the main parameters of soil NO emission (soil temperature and moisture), soil NO probably plays a non-negligible role. For example, soil NO contribute to modelled ozone-temperature relationship at similar degree with BVOC emission and dry deposition (Porter and Heald, 2019). Most previous literature on similar topic do not directly explore the role of such effect, so it is understandable that the authors might not be aware of, and/or the modelling system cannot account for such effects. Yet with recent scientific discussion and development, this paper would benefit greatly from discussing the potential role of soil NO in $O_3$-heat and $O_3$-drought relationships, and how it might affect the conclusion of this paper.

Non-stomatal ozone uptake: Another relevant issue that might have been under the radar of most atmosphere-biosphere chemistry modelers is the importance of temperature- (e.g. in-canopy gas-phase ozonolysis) or water-dependency (e.g. soil, leaf cuticle) of non-stomatal uptake, which is especially frequently highlighted over the Mediterranean region (Fares et al., 2013, 2014; Finco et al., 2018; Gerosa et al., 2005, 2009). In addition, recent study has explicitly shown the importance of accounting for changes in various non-stomatal sinks during heat and drought (e.g. Wong et al., 2022). The EMEP dry deposition scheme does not account for most of these changes. Again this is methodologically acceptable given current state of dry deposition schemes within regional chemical transport models, but 1-2 extra paragraphs should be dedicated to discuss how these changes could affect the result and conclusion.

Minor issues:

L17: "favorable weather condition" for what? Please clarify.

L65 – 66: How exactly do droughts affect land ecosystems more than heatwaves? This need more clarification and references.

L 88: "Dry deposition velocity directly depends on the stomatal conductance…" Non-stomatal uptake is often as important as stomatal in the region of study. "Directly dependent " is a bit too strong wording.

L102: I think the term "agricultural drought" is much less intuitive than simply saying "soil dryness". But I will leave the decision to the authors.

L156 – 157: "ORCHIDEE is composed of three modules. SECHIBA simulates the water and energy cycle. STOMATE resolves the processes of the carbon cycle, allowing an interactive phenology…" ORCHIDEE is likely to provide better $g_s$ than EMEP. Explain why it is not used.

L 251: How much does biomass burning vary with heat and drought in the domain of study? Would that affect the result and conclusion?

L 258: "…and the bulk non-stomatal conductance…" Need much more clarification about the non-stomatal parameterization. Would any part of the total non-stomatal conductance vary with relevant factors like humidity and LAI?

L 300: Explain the cluster approach in more detail.

L 333 – 334: "Even if such validation scores are close to those found in the scientific literature (e.g. Panthou et al., 2018), the temperature uncertainties significantly contribute to those of the $O_3$ simulated by CHIMERE." Especially given a lot of the biospheric parameterizations has non-linear dependence on temperature. Elaborate and analyze the impact on your conclusion.

L 340 – 341: Is such LAI difference applied to the simulations? If not, this statement is confusing and need clarification.

Figure 5: Does the daily mean include only the daylight or all 24 hours? For all day (even daytime) average $1.5 – 2$ cm s$^{-1}$ looks really high comparing to observations over Mediterranean forests. This need a bit more explanation and exploration.

L 456 – 457: Please explain how $\gamma_{SWS_{fit}}$ performs the best since it is not obvious from Fig. 4.

---

## Author Comment (AC1)

> *Reply to the reviewer's comments on the manuscript "Interactions between the terrestrial biosphere and atmosphere during droughts and heatwaves: impact on surface ozone over Southwestern Europe".*
>
> *Guion et al.*

The authors would like to thank the reviewers for their careful reading of the paper and for their comments that improved the quality of the manuscript. All comments have been addressed and a point-by-point answer is provided in the following (in blue after the corresponding comment). The line numbers given in response to comments correspond to the latest version submitted. Finally, modifications made in the new manuscript version are highlighted in the track-changes file provided by the authors.

**Reviewer #1**

This manuscript studies the biospheric impacts of heatwave and drought on surface ozone over southwestern Europe, using well-established numerical models and methodologies. As chemical transport models cannot fully resolve the biosphere, omitting some relevant effects and processes is acceptable, provided that their potential impacts on the results and implications are well discussed. This part is somewhat missing in this manuscript and should be addressed.

*Major issues:*

A) Soil NO emission: Increasing evidence shows that the temperature dependence of soil NO emission is an important part of $O_3$ production, especially in rural and agricultural regions (Oikawa et al., 2015; Romer et al., 2018; Sha et al., 2021), which cover a significant portion of the studies domain. Since this study focus on heatwaves and droughts, which directly affect two of the main parameters of soil NO emission (soil temperature and moisture), soil NO probably plays a non-negligible role. For example, soil NO contribute to modelled ozone-temperature relationship at similar degree with BVOC emission and dry deposition (Porter and Heald, 2019). Most previous literature on similar topic do not directly explore the role of such effect, so it is understandable that the authors might not be aware of, and/or the modelling system cannot account for such effects. Yet with recent scientific discussion and development, this paper would benefit greatly from discussing the potential role of soil NO in $O_3$-heat and $O_3$-drought relationships, and how it might affect the conclusion of this paper.

As the reviewer rightly pointed out, the role of soil NO emissions in ozone production was not developed in the manuscript. Due to the important relationship between temperature, soil moisture and soil NO emissions, heatwaves and droughts probably play an important role in ozone variations.

Although this research focuses on the role of biogenic emissions and dry deposition during extreme events, we agree that it seems important to clarify several points regarding the role of soil NO emissions in the manuscript.

Firstly, we explain in more detail the method used to calculate NO emissions in our simulations. Please see line numbers (Ln.) 283 – 286 in Sect. 3.2.2 "Soil NO emissions are also included in the different simulations. Emission factors are based on the European soil emission inventory (Stohl et al., 1996) which includes both contribution from forests and agricultural soils. Due to the strong temperature dependence, activity factors of soil NO are processed as for biogenic emissions in the MEGAN model. However, no dependence on soil moisture was parameterized.".

Secondly, we discuss the importance of soil NO emissions in ozone production during droughts and heatwaves in the Section 6. Please see Ln. 677 – 681 "In addition, the critical role of soil NO emissions in $O_3$ production is increasingly studied (e.g. Romer et al., 2018), especially in rural areas (e.g. Sha et al., 2021). As hydro-climatic conditions are critical for soil NO emissions, the $O_3$ budget during droughts and heatwaves is likely to be significantly influenced by soil NO emissions. In this study, dependence to soil dryness for NO emissions is not included. Emission pulses can occur when rain follows a drought and emission factors are higher with dry soil than with wet soil (Steinkamp and Lawrence, 2011; Weng et al., 2020).".

B) Non-stomatal ozone uptake: Another relevant issue that might have been under the radar of most atmosphere-biosphere chemistry modelers is the importance of temperature- (e.g. in-canopy gas phase ozonolysis) or water-dependency (e.g. soil, leaf cuticle) of non-stomatal uptake, which is especially frequently highlighted over the Mediterranean region (Fares et al., 2013, 2014; Finco et al., 2018; Gerosa et al., 2005, 2009). In addition, recent study has explicitly shown the importance of accounting for changes in various non-stomatal sinks during heat and drought (e.g. Wong et al., 2022). The EMEP dry deposition scheme does not account for most of these changes. Again, this is methodologically acceptable given current state of dry deposition schemes within regional chemical transport models, but 1-2 extra paragraphs should be dedicated to discuss how these changes could affect the result and conclusion.

We thank the reviewer for this comment. We agree that non-stomatal conductance should be discussed further due to its importance in the budget and its relationship with meteorological conditions. Please see Ln. 682 – 691 "Some recent knowledge on fundamental processes that allow a better representation of surface-atmosphere interactions during extreme weather events is not yet integrated by much, if not all, of the modelling community. Among the many examples are the increased emission of monoterpenes and sesquiterpenes during the development of droughts (e.g. Bonn et al., 2019; Peron et al., 2021) or the in-canopy chemistry that is ignored or approximated by the "big leaf" model approach (e.g. Clifton et al., 2020a). Several studies show that non-stomatal conductance in the Mediterranean counts as much as stomatal conductance in the $O_3$ sink budget (e.g. Gerosa et al., 2009; Sun et al., 2022). Moreover, non-stomatal conductance also seems to be significantly affected during droughts and heatwaves. In contrast to stomatal conductance, droughts and heatwaves

should have opposite effects: an increase in non-stomatal conductance during heatwaves and a decrease during droughts (Wong et al., 2022). However, such changes are not taken into account in the EMEP deposition scheme implemented in CHIMERE. There is a real need for a better representation of stomatal conductance in deposition schemes.".

In addition, further information on the parameterization of non-stomatal conductance used in this research is provided in response to comment G of the minor issues (Reviewer #1). Please see Ln. 300 – 304 "$G_{ns}$ depends on three resistances: the external leaf uptake, the ground surface and the in-canopy. The external leaf uptake resistance is fixed at 2500s/m, the ground surface resistance is estimated from tabulated values (PFT specific) and the in-canopy resistance varies with the surface area index which is expressed in terms of LAI. The LAI used in the dry deposition scheme is parameterized (with no inter-annual variation). Information on phenology and biomass variation for each land cover type was collected from several studies in Emberson et al. (2000).".

*Minor issues:*

    A) L17: "favorable weather conditions" for what? Please clarify.

We thank the reviewer for this comment. I meant favorable weather conditions for photochemistry.
Please see Ln. 18 – 19 "The overall increase of surface $O_3$ during both heatwaves and droughts would be explained by $O_3$ precursor emission enhancement, $O_3$ dry deposition decrease and favorable weather conditions for photochemistry.".

    B) L65 – 66: How exactly do droughts affect land ecosystems more than heatwaves? This needs more clarification and references.

Droughts (mainly by water stress) can lead to a decrease and even a stop of photosynthetic activity. Satellite data in Guion et al. (2021) show that droughts lead to an average decrease in biomass of -10%. Heat stress during heatwaves induces a smaller decrease (-3% on average). Moreover, depending on the season and latitude, heatwaves can increase plant activity and lead to an increase in biomass. (Baumbach et al., 2017).
The words "and to a larger extent than heatwaves" may be confusing for the reader. For the sake of clarity and length, we delete this part of the sentence. In addition, we add relevant reference.
Please see Ln. 66 – 67 "Another difficulty is that droughts affect not only the atmosphere but also the land biosphere through soil dryness and plant activity decline (e.g. Vicente-Serrano, 2007; Vicente-Serrano et al., 2013).".

C) L88: "Dry deposition velocity directly depends on the stomatal conductance…" Non-stomatal uptake is often as important as stomatal in the region of study. "Directly dependent" is a bit too strong wording.

We agree with the reviewer that this statement may be too strong wording. We modified it. Please see Ln. 92 – 93 "Dry deposition velocity in the Mediterranean depends on both stomatal and non-stomatal conductance (Lin et al., 2020; Sun et al., 2022).".
In addition, the importance of non-stomatal uptake and how it is discussed in the new version of the manuscript are reported in comment B of the major issues (Reviewer #1).

D) L102: I think the term "agricultural drought" is much less intuitive than simply saying "soil dryness". But I will leave the decision to the authors.

Because agricultural drought as a type of drought has already been defined before in the text (and largely used afterwards), we prefer to keep the sentence as it is. Nevertheless, we have added the words for clarity "soil dryness" to the following sentence "Meteorological droughts correspond to a rainfall deficit or an excess of evapotranspiration, agricultural ones to soil water shortage for plant growth (soil dryness), and hydrological ones to surface and/or underground flow decrease." (Ln 64 - 66).

E) L156 – 157: "ORCHIDEE is composed of three modules. SECHIBA simulates the water and energy cycle. STOMATE resolves the processes of the carbon cycle, allowing an interactive phenology…" ORCHIDEE is likely to provide better gs than EMEP. Explain why it is not used.

Although experiments to couple CHIMERE and ORCHIDEE have been performed (e.g. Anav et al., 2011; Franz et al., 2017), they focused on $O_3$ uptake and plant damage. So far, there is no complete coupling between the two models considering the dry deposition of all gases necessary for an accurate calculation of ozone budget (ozone, ozone precursors, and ozone depleting gases). Such work, which would be very interesting to carry out, does not fall within the objectives of this paper.

F) L251: How much does biomass burning vary with heat and drought in the domain of study? Would that affect the result and conclusion?

Indeed, wildfire activity increases during droughts and heatwaves. The enhancement of the wildfire activity during simultaneous droughts and heatwaves is quantified by a burned area and a fire intensity 2.1 and 2.9 times larger than for wildfires under normal conditions (Guion et al., 2021). In this research, we calculated that biomass burning CO emissions are 13 times larger during combined heatwaves and droughts compared to normal days.
However, the significant contribution of fire emissions to ozone pollution occurs at local scale and over a duration of several days (e.g. Northern Portugal over summer 2013). We performed CHIMERE simulations without fire emissions (summer 2012 and 2013). For reasons of length and clarity, we have not

mentioned that in the manuscript. The results and conclusion of this research remain the same.

G) L258: "...and the bulk non-stomatal conductance..." Need much more clarification about the non-stomatal parameterization. Would any part of the total non-stomatal conductance vary with relevant factors like humidity and LAI?

The EMEP dry deposition scheme implemented in CHIMERE includes stomatal and non-stomatal conductance. Non-stomatal conductance depends on the in-canopy resistance, the external leaf uptake resistance and the ground surface resistance. In-canopy resistance varies with the Surface Area Index (SAI) which is defined in terms of LAI; SAI is set to the LAI value within the growing season and 1 outside the growing season. Therefore, the effect of droughts that are implemented on the biomass, also modify the non-stomatal conductance. Regarding the external leaf uptake resistance, relative humidity plays a role (e.g. Erisman et al., 1994). However, this resistance is approximated here by using a constant value from Emberson et al. (2000).

Further information on the parameterization of non-stomatal conductance is provided in Sect. 3.2.3. Please see Ln. 289 – 290 "Canopy resistance is calculated from stomatal conductance ($g_{sto}$) which increases proportionally with LAI, and the bulk non-stomatal conductance ($G_{ns}$)." and Ln. 300 – 304 "$G_{ns}$ depends on three resistances: the external leaf uptake, the ground surface and the in-canopy. The external leaf uptake resistance is fixed at 2500s/m, the ground surface resistance is estimated from tabulated values (PFT specific) and the in-canopy resistance varies with the surface area index which is expressed in terms of LAI. The LAI used in the dry deposition scheme is parameterized (with no inter-annual variation). Information on phenology and biomass variation for each land cover type was collected from several studies in Emberson et al. (2000).".

H) L300: Explain the cluster approach in more detail.

We agree that although we have explained that droughts and heatwaves are identified from the PLA indicator, the definition of clusters is not clear.

As described in Sect. 3.1.2 - Indicators of drought and heatwave, "A heatwave/drought is identified when the daily $PLA_{T2m/SD}$ is positive for three consecutive days." (Ln. 195 – 196). Normal conditions are present when there is neither drought nor heatwave event. Extreme weather events can be isolated or combined, as shown in Figure A.

We have added the following sentences in the manuscript. Please see Ln. 510 – 514 "Clusters of droughts and heatwaves (isolated or combined) are constructed based on the $PLA_{T2m}$ and $PLA_{SD}$ indicators, allowing to analyze the statistical variation of $C_5H_8$ emissions, $O_3$ stomatal conductance and $O_3$ surface concentration (both from observations and simulations). Using those indicators, the following conditions were defined and grouped into clusters: "heatwaves or droughts", "heatwaves and droughts", "heatwaves and not droughts" and "droughts and not heatwaves". Normal conditions are defined as no drought and no heatwave.".

[Figure]

Figure A - Description of extreme weather event clusters

I) L333 – 334: "Even if such validation scores are close to those found in the scientific literature (e.g. Panthou et al., 2018), the temperature uncertainties significantly contribute to those of the $O_3$ simulated by CHIMERE." Especially given a lot of the biospheric parameterizations has nonlinear dependence on temperature. Elaborate and analyze the impact on your conclusion.

We thank the reviewer for this comment. We address this comment in the conclusions. Please see Ln. 670 - 675 "Important uncertainties appear to be related to BVOC emissions (especially due to the land cover classification), to $NO_x$ concentrations for which CHIMERE presents limited performance scores of validation, to $O_3$ deposition and finally to meteorological conditions (e.g. temperature and PBL height). For instance, the simulated daily maximum temperature that is underestimated in the Northern Mediterranean compared to observations (see Sect. 4), may induced a decrease of $O_3$ precursors, especially as the emissions–temperature relationship of BVOCs is exponential, with an optimal temperature for $C_5H_8$ species (Guenther et al., 1993).".

J) L340 – 341: Is such LAI difference applied to the simulations? If not, this statement is confusing and need clarification.

Indeed, such LAI difference is applied to the simulations conducted for the sensitivity analysis of $C_5H_8$ emissions to soil dryness and biomass decrease effects (Sect. 4.1.1.).
Such information is given in the explanation of the different conducted experiences. Please see Ln. 319 – 322, in Sect. 3.3 Experiments "The LAI used in the emission scheme of MEGAN is year dependent (MODIS observations). To evaluate the effect of biomass decrease by droughts, a simulation with LAI corresponding to a wet summer was used ("HighLAI-emiss"). Summer 2012 which was affected by an important biomass decrease over most of the study area (SI: Fig. S3), has been simulated with the LAI of the wet summer of 2014 (higher than the 2012-2014 mean).".

K) Figure 5: Does the daily mean include only the daylight or all 24 hours? For all day (even daytime) average 1.5 – 2 cm s-1 looks really high comparing to observations over Mediterranean forests. This needs a bit more explanation and exploration.

We thank the reviewer for his comment. The daily average includes all 24 hours. We agree that in comparison with observations, such a range of values seems to be high. In order to explore and discuss these results, we compared the dry ozone deposition with the observational data available for summer 2012 at the Casterlporziano2 station in Italy (Figure B). A comparative analysis including several additional references is presented in Section 5.1.2. Please see Ln. 458 – 469 "To the best of our knowledge, there is no study yet in the scientific literature that fully assess the $O_3$ dry deposition of CHIMERE against observations or its sensitivity to different meteorological forcings. Therefore, we have compared our results to the measured data available for summer 2012 at the Castelporziano station from the European Fluxex Database (Fig. 6). Based on the PLA indicator, the Lazio region was affected by a severe drought all along the summer (mean $PLA_{SD}$ of +0.05). All simulated experiments overestimate $O_3$ deposition compared to observations. The "LAIdecr-dep/f SW S -dep" experiment has the smallest average bias (-0.19e-6 $g/cm^2$) and the highest correlation (R coefficient of 0.40). Such overestimation has also been calculated for models whose gaz deposition scheme is based on Wesely (1989) "big leaf" parameterization (e.g. Michou et al., 2005; Huang et al., 2022). As the canopy conductance increases proportionally with the prescribed LAI (Emberson et al., 2000), this could be explained by an overestimation of the LAI that is almost two times larger than the mean LAI reconstructed from MODIS over this area. The importance of representing processes dynamically (as opposed to fixed parameters, especially for the non-stomatal conductance) is also highlighted in order to better simulate the diurnal deposition cycle, and so the daily average values (Huang et al., 2022).".

[Figure]

Figure B - Observed $O_3$ deposition flux [g/cm$^2$] during the summer 2012 at the Castelporziano station (IT-Cp2, Lazio region) from the Europe Flux database, compared to the different simulated experiments undertaken by the CHIMERE model. Maximum $PLA_{T2m}$ and $PLA_{SD}$ are +3.88°C and +0.09 of soil dryness index respectively.

L) L456 – 457: Please explain how $\gamma SWSfit$ performs the best since it is not obvious from Fig. 4.

As the reviewer correctly mentioned, $\gamma SWSfit$-emiss is not the experiment that performs the best all along the summer. Please see Ln. 425 – 428 "Averaged over the summer, the lowest mean bias with the observations is obtained with "$\gamma SWSfit$-emiss" (-28 pmol/mol) and the largest one with "$\gamma SM$-emiss" (-151pmol/mol). Over July and August, "$\gamma SWS$-emiss" experiment presents the lowest mean bias (+56 pmol/mol).".

However, our choice to use the $\gamma SWSfit$-emiss experiment for the simulations is not only based on the score performance at one specific station. We chose $\gamma SWSfit$ because it remains rather conservative, not in the lower limit of the $C_5H_8$ reduction range. For instance, "$\gamma SWS$-emiss" induces a strong reduction (up to -95% on average) over the whole southwestern part of the area, as shown in Figure 2 of the manuscript.

We justified it in Ln. 507 – 509 "The "all-emiss-dep" experiment has been chosen because it includes drought and heatwave effects in the most comprehensive way. Moreover, the $C_5H_8$ emission approach "γSWSfit-emiss" has shown good performance compared to observations (Fig. 4), remaining rather conservative, not in the higher limit of the $C_5H_8$ reduction range".

**Reviewer #3**

This manuscript focuses on describing WRF/CHIMERE simulations covering SW Europe in summer 2012-2014. In these simulations, the responses of MEGANv2.1 biogenic VOC emissions and dry deposition to heat and water stresses as well as biomass changes were represented differently. The study compares $O_3$ mass concentrations and exceedances, HCHO columns, biogenic isoprene emissions, dry deposition velocity and $O_3$ stomatal conductance from these various simulations during heatwaves, isolated droughts, combined droughts, and normal periods. In-situ observations of 2m temperature, $O_3$, and isoprene and satellite (OMI) HCHO column data were used for model evaluation.

This study addresses a topic that has become increasingly popular, and their modeling tools have not been used in previous works to address such a topic. More efforts were devoted to modifying the biogenic emission scheme. The paper falls within the scope of ACP/EGUsphere. Please see below my comments.

*Paper structure:*

A) Introduction contains information that is not directly relevant to what this work addresses and should belong to discussions on the limitation of this work in the end, for example, monoterpene emissions, and the impacts of the drought phases on biogenic emissions of various species.

We thank the reviewer for this comment. We have taken it into account and made some changes. Information on the limitations of models to represent BVOC emissions and gaz dry deposition during droughts is moved into the discussion (please see Ln. 681 – 691).

B) Section 2 should include some information on $NO_2$ observations (chemiluminescence analyzer?) and their uncertainty used for model evaluation. Please consider moving L321 here.

Indeed, the use of $NO_2$ measurements for model validation is not clearly indicated in the presentation of the observations. Please see Ln. 127 "The in-situ measurements of surface $O_3$ and $NO_2$ provided by the EEA are used." and Ln. 133 – 137 "Thunis et al. (2013) quantify the various sources of uncertainty for $O_3$ measurements (e.g. linear calibration and ultraviolet photometry) and estimated a total uncertainty of 15%, regardless of concentration level. There are also considerable uncertainties in the measurements of $NO_2$. Lamsal et al. (2008) emphasize that the chemiluminescence analyzer, the measurement technique primarily found in air quality stations, is subject to significant interference from other reactive species containing oxidized nitrogen (e.g. PAN, $HNO_3$). This can lead to an overestimation of measured $NO_2$ concentrations.".

C) Section 2 should contain more information about isoprene observations from "different experiments" (methods, uncertainty) rather than simply providing a link. It seems that data from only one station (Ersa) were used? Can the location of this station also be indicated in a map?

As correctly pointed out by the reviewer, information on the $C_5H_8$ concentration measurement station is missing. Furthermore, we have added information on the $O_3$ deposition flux measuring station we added (see comment K, minor issues, reviewer #1). The use of "different experiments" referred to the modelling experiments. We have clarified this point. Please see Ln. 162 – 170 "Secondly, flux measurements are used for comparison with the different modelling experiments carried out to analyze the sensitivity of biogenic $C_5H_8$ emissions and $O_3$ deposition to the effects of biomass decrease and soil dryness. However, there are few flux measurements available that cover at least several weeks during summer 2012. The ERSA station (FR0033R), located at Cape Corsica in France (42.97°N, 9.38°E) is used to assess surface concentration of $C_5H_8$ measured by a steel canister instrument at 4.0 meters above the surface. The data are provided by the EBAS infrastructure (https://ebas.nilu.no/). Dry deposition flux measurements of $O_3$ are also used at one station. This is the Castelporziano station (IT-Cp2) located in the Lazio region of Italy (41.70°N, 12.36°E) measuring at 14.9 meters above the surface (eddy covariance technique). A full description of the measurement data is provided in Fares et al. (2013). The data were downloaded from the European Fluxes Database Cluster (EFDC, http://www.europe-fluxdata.eu/).". Measurement uncertainty (if available) is displayed on the time series.
Finally, as the maps in the manuscript are already well filled, we assume that the indication of the ID, coordinates and region of the stations represents sufficient information for the readers.

D) The authors should clarify that the E-OBS dataset mentioned in Section 2 (Cornes et al.) is a 0.25-degree gridded product which was regridded for model evaluation shown in Figure S5 (please confirm).

We thank the reviewer for this clarification. We have indeed added this information to the text. Please see Ln. 158 - 161 "Firstly, in-situ observations of temperature at 2 meters above the surface ($T_{2m}$) from the E-OBS data set (Cornes et al., 2018) are used to validate the simulated temperature by the Weather Research and Forecasting (WRF) model (Skamarock et al., 2008). The E-OBS gridded product with a resolution of 0.25° × 0.25° was regridded to the Med-CORDEX domain (see Sect. 3.1.1).".

E) Section 3.2, descriptions on MEGAN scheme (a process in CHIMERE) could better be merged into 3.3.

Indeed, since the MEGAN model is used as a processed scheme in CHIMERE, we have included "3.2.2. MEGAN and the soil moisture factor" in the Subsection "3.2 CHIMERE".

F) Why isn't Section 3.3.4 (model validation results) a part of Section 4 (results)?

We have created a specific section for model validation (Section 4. Validation of surface $O_3$, $NO_2$ and $T_{2m}$). Indeed, it seemed partially wrong to leave this part as a sub-section of "3. Models", as the observations are used for validation. However,

we decided not to include it in the section "5. Results" as it does not reflect the main objectives of our study, where results are already divided into three main parts ((i) Sensitivity to soil dryness and biomass decrease effects, (ii) statistical variation during droughts and heatwaves and (iii) threshold level exceedance of $O_3$).

*Modeling and analysis approach:*

A) Section 3.1.1: Physics schemes used in WRF simulations should be specified. How were these simulations initialized for atmosphere and land? Please also clarify: what was the reanalysis data used as lateral boundary conditions, ERA-Interim (L165) or a NCEP product (L627)? These can all strongly affect your WRF results. See: Huang et al. (https://doi.org/10.5194/gmd-10-3085-2017).

We thank the reviewer for this comment. One important point needs to be clarified. The simulations performed with the WRF weather model in WRF-CHIMERE and WRF-ORCHIDEE are not the same. The configurations are different. WRF-ORCHIDEE simulations from Guion et al. (2021) are used for the calculation of drought and heatwave indicators and the SWS (used in the $\gamma_{SWSfit}$ experiment, see section 3.3.3). For those simulations, the ERA-Interim reanalysis were used as lateral boundaries conditions. A full description of the configuration used for WRF with ORCHIDEE is presented in Guion et al. (2021): "The ARW (Advanced Research WRF) non-hydrostatic dynamical core was selected together with a set of physics packages appropriate for resolutions of about 20 km. These include in particular the single-moment 5 class microphysics scheme (Hong et al. 2004), which produces the clouds and their properties in interaction with the radiation scheme developed by Iacono et al. (2008). Convection is parametrized at these resolutions with the Kain–Fritsch scheme (Kain 2004) and the shallow convection scheme proposed by Park and Bretherton (2009). The interaction with the surface occurs through the Mellor–Yamada Level-3 representation of boundary layer turbulence, developed by Nakanishi and Niino (2009).".
For reasons of clarity and length, we prefer not to add such a description to the manuscript but rather refer the reader to Guion et al. (2021). Please see Ln. 181 - 182 "The full description of the configuration used with RegIPSL is presented in Guion et al. (2021).".
We agree with the reviewer that information on the configuration of the WRF model used with CHIMERE is missing in the manuscript. Please see Ln. 225 - 230 "The WRF model used with CHIMERE, which has a different configuration from the RegIPSL model (Guion et al., 2021), has 15 vertical layers, from 998hPa up to 300hPa. The physics interface for the surface is provided by the Noah Land Surface Model. The aerosol-aware Thompson scheme (Thompson and Eidhammer, 2014) is used for the mycrophysics parameterization. Horizontal and vertical advection are based on the scheme of Van Leer (1977). The reanalysis meteorological data for the initial and boundary conditions are provided by National Centers of Environmental Predictions (NCEP). The physical and chemical time steps are respectively 30 and 10 minutes."

B) I am confused about the vertical layers of WRF/CHIMERE, 46 layers (L164) or 15 layers (L241)?

This comment is related to the previous one (Modeling and analysis approach, comment A, reviewer #3). The configuration of the WRF model used with ORCHIDEE is not the same as with CHIMERE. There are 46 vertical levels in the WRF model used with ORCHIDEE and 15 when used with CHIMERE.
For the sake of clarity, we have removed this information on the WRF-ORCHIDEE simulations and referred to Guion et al. (2021) where a full description of the WRF-ORCHIDEE simulation setup is provided.

C) WRF-Noah has been mentioned a few times, but it is not clear enough how this can be directly comparable with WRF-ORCHIDEE based analysis because Noah and ORCHIDEE are quite different land models and they may yield quite different soil moisture fields. In the gamma *SM-emiss* experiment why couldn't soil moisture from ORCHIDEE be used instead of Noah?

WRF-ORCHIDEE and WRF-Noah are indeed different models with their own representation of soil and vegetation processes leading to a different water balance. Consequently, the soil moisture fields are different, with their own amplitude and mean values. The experiments presented in this paper do not compare soil moisture, but rather two plant water stress activity factors calculated from the outputs of WRF-ORCHIDEE and WRF-Noah.
The soil moisture activity factor needs to be parameterized from the output values (Muller et al., 2007; Wang et al., 2021). For instance, the WRF-ORCHIDEE soil moisture index has a range of variability of about 0 to 1. It cannot be used in the "*SM-emiss*" experiment where the wilting point is defined for the WRF-NOAH soil moisture index which varies between 0 and 0.5 (depending on vertical levels). This is also the reason why we used the output of WRF-ORCHIDEE in the function of Bonn et al. (2019) which was based on a soil water availability index (between 0 and 1).

D) I found that the busy box plots in Section 4 are slightly difficult to follow as the categories have overlaps, not clearly linking their results with the phases/severity of droughts and heatwaves, nor are they directly linked with previously shown time series plots.

The box plots correspond to the clusters of extreme weather events used for the analysis in Section 5.2. As requested by reviewer #1 (comment H, minor issues) to provide more information to the reader, we have explained the cluster approach in more detail. Please see Ln. 510 – 514 "Clusters of droughts and heatwaves (isolated or combined) are constructed based on the $PLA_{T2m}$ and $PLA_{SD}$ indicators, allowing to analyze the statistical variation of $C_5H_8$ emissions, $O_3$ stomatal conductance and $O_3$ surface concentration (both from observations and simulations). Using those indicators, the following conditions were defined and grouped into clusters: "heatwaves or droughts", "heatwaves and droughts", "heatwaves and not droughts" and "droughts and not heatwaves". Normal conditions are defined as no drought and no heatwave.".

The droughts and heatwaves presented in the time series are based on the same indicator as for the construction of the clusters.

    E) Figure S1: soil type USGS? Please double check. USGS seems to be the source of the land cover input.

As pointed out by the reviewer, the USGS is the source of the land cover input but also of the soil type. Tabulated values of the wilting point determined by soil type (from Chen and Dudhia (2001)) were spatialized using the soil type map provided by USGS. Soil is represented by relative percentages of sand, silt and clay for each model cell. The USGS database, called STATSGO-FAO, is used and 19 different soil types are recorded in the global database (available at http://soils.usda.gov/survey/geography/). Please see Ln. 269 – 272 "$\theta_w$ values are computed over the domain (Supplementary Information: Fig. S1) using tabulated values from Chen and Dudhia (2001) that are soil type specific and parameterized on Noah soil wetness values. These values are spatialized using the soil texture map provided by the USGS (STATSGO-FAO product)".

    F) In Section 3.3.1, please consider discussing the drought impacts on biomass burning emissions because this pathway also affects $O_3$ variability. Was soil NO emission included, and if so, was it from MEGAN? Also, the model chemical initial/boundary conditions and their quality should be mentioned. The model errors due to chemical initial/boundary conditions during stagnant and dynamic atmospheric conditions, which have connections with droughts and heatwaves, may be different.

The first part of this comment, about biomass burning emissions, is addressed in our reply to comment F (minor issues, Reviewer #1). However, we have added a sentence about it in the Introduction. Please see Ln. 77 – 79 "Finally, large amount of $O_3$ precursors emitted during biomass burning enhanced by droughts and heatwaves, can contribute to $O_3$ pollution peaks (e.g. Hodnebrog et al., 2012)".
The second part of this comment, about NO emissions, is also addressed in a response to Reviewer #1 (major issues, comment A).
Finally, information and references about chemical initial and boundary conditions are provided in Ln. 236 – 237 "Chemical boundary conditions for the larger domain are provided by a climatology from the global CTM LMDZ4-INCA3 (Hauglustaine et al., 2014) for trace gases and non-dust aerosols, and from the GOCART model (Chin et al., 2002) for dust.". As these are nested simulations, we use the outputs of the larger domain model as boundary conditions for the smaller domain. Finally, we have a spin-up of 5 days (before the analyzed period) in order to limit the impact of initial conditions on the stability of the simulation. Please see Ln. 231 – 232 "The simulations have been performed for June-July-August (with a 5 days spin-up) of years 2012-2013-2014 over two nested simulation domains (Fig. 1).".

G) Section 3.3.2: How about nonstomatal terms, perhaps Wesely? The authors have recognized that there are other approaches in literature (including Lin et al. and Clifton et al. that have already been cited) to represent soil moisture and vegetation impacts on dry deposition, as well as its stomatal and nonstomatal terms. They should point out (e.g. at L410-415, when comparing their results with Lin et al.) that the choice of the dry deposition scheme can strongly affect one's findings. The authors may like to be aware that, based on the dry deposition scheme used here, Anav et al. (https://doi.org/10.5194/acp-18-5747-2018) have also quantified the soil moisture impacts on gsto and ozone; and Huang et al. (https://doi.org/10.5194/acp-22-7461-2022) quantified the soil moisture impacts on dry deposition and ozone based on different schemes.

The first part of this comment, about non-stomatal conductance, is addressed in our reply to comment B (major issues, Reviewer #1).
We thank the reviewer for the suggested references in the second part of the comment. We have taken them into account. Please see Ln. 455 – 457 "The sensitivity of dry deposition velocity to soil moisture can be considerably different from one deposition scheme to another. Using the same deposition scheme as in the present study, Anav et al. (2018) calculated an average decrease of 10% over Europe in dry $O_3$ deposition." and Ln. 465 – 469 "As the canopy conductance increases proportionally with the prescribed LAI (Emberson et al., 2000), this could be explained by an overestimation of the LAI that is almost two times larger than the mean LAI reconstructed from MODIS over this area. The importance of representing processes dynamically (as opposed to fixed parameters, especially for the non-stomatal conductance) is also highlighted in order to better simulate the diurnal deposition cycle, and so the daily average values (Huang et al., 2022).".

H) L283-289 and Figure S3: Please clarify the sources of LAI. Specifically, is LAI interannual variability based on ORCHIDEE or some type of satellite data? Does "year dependent" mean summertime averaged or annually averaged for different years? Is the constant LAI used in dry deposition modeling from a climatological product, and if so, what is it? Could the model-based LAI be presented in maps and if possible, be evaluated?

The LAI used in the BVOC emissions and gaz dry deposition scheme comes from two different sources. The LAI used in MEGAN is derived from a satellite product described in the manuscript. Please see Ln. 258 – 261 "γLAI is the activity factor based on LAI observations from the MODIS MOD15A2H product (Myneni et al., 2015) improved by Yuan et al. (2011) (http://globalchange.bnu.edu.cn/research/lai). This improvement is undertaken with a two-step integrated method: (1) the Modified Temporal Spatial Filter is used to fill the gaps and replace low quality data by consistent data; (2) the TIMESAT Salvitzky and Golay filter is applied to smooth the final product. The temporal resolution is 8 days."
By "year dependent", we mean that the reconstructed LAI from satellite instrument varying inter- and intra-annually is used.
Concerning the LAI used in the dry deposition scheme, it is fixed and parameterized for each land cover type. It comes from Emberson et al. (2000) where many of the parameters used to calculate stomatal conductance are also

land cover specific. As presented in Table A, information on phenology and biomass variation for each land cover was collected from several studies.

As suggested by the reviewer, we have compared the two LAI datasets (Figure C). Compared to the MODIS LAI (dry summer of 2012), the LAI from Emberson et al. (2000) is underestimated over the forests (about -1) and overestimated over grass and croplands (about +2). These results may explain the overestimation of the simulated $O_3$ dry deposition flux compared to the observations (Fig. 6 in the manuscript). Please see Ln. 465 – 467 "As the canopy conductance increases proportionally with the prescribed LAI (Emberson et al., 2000), this could be explained by an overestimation of the LAI that is almost two times larger than the mean LAI reconstructed from MODIS over this area.".

| Land-cover Class | SLAI | ELAI | LAI_min | LAI_max | SLAI_len | ELAI_len | References |
|---|---|---|---|---|---|---|---|
| Temp./boreal coniferous forest | 0 | 365 | 3.4 | 4.5 | 192 | 96 | Abdulla & Lettenmaier (1997), Neilson & Marks (1994), Beadle *et al.* (1982) |
| Temp./boreal deciduous forest | =SGS | =EGS | 3.5 | 5 | 56 | 92 | Abdulla & Lettenmaier (1997), *Olson et al.* (1985), Padro (1996), Jakobsen et al. (1996) |
| Medit. needleleaf forests | 0 | 365 | - | 3.5 | - | - | Hassika et al. (1997) |
| Medit. broadleaf forests | 0 | 365 | - | 4.5 | - | - | Sala & Tenhunen (1994), Gratani (1993), Sala & Tenhunen (1996) |
| Temp. crops | =SGS | =EGS | 0 | 5 | 70 | 22 | Olson *et al.* (1985), Boumann (1995) |
| Med. crops | =SGS | =EGS | 0 | 3.5 | 70 | 44 | Studeto et al. (1995), Bouman (1995) |
| Root crops | =SGS | =EGS | 0 | 6 | 35 | 65 | Pers. comm. Vandemeiren (EU CHIP Project) |
| Vineyards | 0 | 365 | - | 3 | - | - | Padro (1994), Mascart *et al.* (1991) |
| Temp. orchard | =SGS | =SGS | 1 | 3.5 | 60 | 100 | Tustin et al. (1988); Palmer et al. (1992) |
| Medit. orchard | 0 | 365 | - | 3 | - | - | Padro (1994), Mascart *et al.* (1991) |
| Grasslands | 0 | 365 | 2 | 5.5 | 140 | 135 | Mascart *et al.* (1991), Olson *et al.* (1985) |
| Moor/heathland | 0 | 365 | 2 | 5 | 192 | 96 | Abdulla & Lettenmaier (1997), Neilson & Marks (1994) |
| Med. scrub | 0 | 365 | - | 4.5 | - | - | Pio et al. (2000) |

Table A – Maximum LAI and course of LAI throughout the growing season for the different land cover categories. Table from Emberson et al. (2000)

[Figure]

Figure C – Summer mean LAI from Emberson et al. (2000) (top panel) and from MODIS MOD15A2H product for 2012 (bottom panel)

I) Figure S7: Definition of chemical regime parameter is not clear. Do low-NO$_x$ regime and high-NO$_x$ regime refer to NO$_x$ limited and NO$_x$ saturated regimes, respectively? What numbers are considered low and high, respectively? It's hard to find such information from the link provided in the figure caption.

Low-NO$_x$ and high-NO$_x$ do not refer to NO$_x$ limited and NO$_x$ saturated regimes. α calculates the ratio of the reaction rate of RO$_2$ radicals with NO (high-NO$_x$ regime) with respect to the sum of reaction rates of the reactions with HO$_2$ and RO$_2$ (low-NO$_x$ regime). In other words, it represents the part of RO$_2$ radicals reaction with NO (representative of high NO$_x$ yields). It gives a relative indication of low-NO$_x$ (low α, about 0.5 in summer average in this study) and high-NO$_x$ (high α, about 0.9) regime areas. Low-NO$_x$ and high-NO$_x$ conditions in terms of VOC/NO$_x$ concentration ratio are detailed in Zhang et al. (2013).

We have made this clear in the legend to the figure. Please see Figure S7 "Daily mean chemistry regime parameter [α] averaged over the summer 2012, 2013 and 2014 ("Reference" simulations). α calculates the ratio of the reaction rate of $RO_2$ radicals with NO (high-$NO_x$ regime) with respect to the sum of reaction rates of the reactions with $HO_2$ and $RO_2$ (low-$NO_x$ regime). It gives a relative indication of low-$NO_x$ (low α, about 0.5) and high-$NO_x$ (high α, about 0.9) regime areas that are detailed in Zhang et al. (2013). More information about the calculation method of α is provided on the online documentation (https://www.lmd.polytechnique.fr/chimere/)."

   J) L425: "Surface $O_3$ remains high above the sea due to transport and the absence of dry deposition" - please confirm the (lack of) treatment of dry deposition over the water. How does this contribute to modeled $O_3$ errors over land via sea-land breezes and large-scale flows?

We thank the reviewer for this comment. We should point out that even though there is no canopy conductance over water, there is still dry deposition (due to the other resistances such as the aerodynamics one). A land-sea mask was applied to the map in the Figure 5 when it should not be. We have modified this (please see Fig. 5 In the manuscript). The dry deposition velocity over water is almost constant, about 0.18 cm/s. The transport of $O_3$ over water is also considered. Finally, we have changed this sentence. Please see Ln. 478 – 479 "Surface $O_3$ remains high above the sea due to transport and the absence of deposition in the canopy (e.g. 120µg/m$^3$ over the Adriatic sea)".

*Uncertainty associated with their results and conclusions:*

   A) As the authors acknowledged, uncertainty due to the outdated land cover and soil type (along with soil-type-dependent wilting point) inputs reduces the robustness of their results and that updated versions of input data are available. Many studies have assessed the impacts of land cover inputs (and LAI) on MEGAN biogenic emissions which may be cited. In the supplement, it'd be helpful to show a land cover input (of WRF/ORCHIDEE and MEGAN) map in comparison with MODIS to help understand the statements at L495-500.

As suggested by the reviewer, we have added references to this topic to complete our discussion. Please see Ln. 556 – 559 "The choice as well as the temporal evolution of the land cover database are crucial for the calculation of $C_5H_8$ emissions (Chen et al., 2018). Despite the use of satellite data for land cover, significant uncertainties remain in the calculation of $C_5H_8$ emissions due to the classification of vegetation types and species (Opacka et al., 2021).".
In addition, we have included in supplement material the land fraction maps for the vegetation types used CHIMERE and those identified by MODIS (Figure D). This illustrates the findings in Ln. 553 – 556 "After aggregation of the USGS land cover classes, the vegetation type assumed in CHIMERE in the Balkans, for instance, is 57% of forest cover, 9% grassland and 33% cropland. Using the MODIS MCD12 product (Friedl et al., 2010), we find a different distribution, with 30% of forest cover, 25% grassland and 31% cropland (SI: Fig. S10).". Please

note that this difference in land coverage also applies to regions other than the Balkans.

[Figure]

Figure D – Land cover fraction of cropland, grassland and forests over the Southwestern Europe from USGS (left column) and MODIS MCD12 product (right column).

B) PLA(T2 and SD) is developed based on model results which are uncertain. While model absolute T2 values (which are not shown in maps) are evaluated with E-OBS, the performance is hard to be directly linked to PLA(T2). Also could PLA(T2 and SD) be evaluated against, for example, independent, widely used drought indicators/indexes (see https://doi.org/10.1175/BAMS-D-20-0087.1 for some examples)? How may the uncertainty in your drought/heatwave classifications affect the conclusions?

We understand the concern about how the biases in both temperature and rainfall are considered in the extreme event definition. To check the robustness of our analysis, heatwaves were also identified based on the observed 2m above surface temperature from the E-OBS datasets. The spatial and temporal variations of heatwaves are similar between observations and simulations (Figure E & F). As the bias between simulated and observed $T_{2m}$ is constant and the correlation coefficient high (Guion et al., 2021), the variability and magnitude of the anomalies from the percentile 75 are close between E-OBS and Med-CORDEX. However, even if peak occurrence is well represented in Med-CORDEX, we noticed that simulated heatwaves are slightly more intense (no more than +0.50°C) than the observed ones. It suggests that temperature peaks in Med-CORDEX can be slightly larger than in E-OBS. Computing the mean bias of heatwave characteristics between Med-CORDEX and E-OBS, we found +0.16°C for the intensity, +0.07 for the fraction of days and +0.03 days for the longest extreme event.

The simulated precipitation is also characterized by a constant bias and high correlation with the observed precipitation from E-OBS (Guion et al., 2021). The variables needed for drought detection are soil dryness (for PLA method), temperature and potential evapotranspiration (for SPEI, another drought index). Such type of observations are too sparse for covering the whole Western Mediterranean and are very uncertain. Therefore, we calculated the SPEI based on Med-CORDEX outputs to compare it with the PLA indicator over sub-regions of the Mediterranean (Figure G). We found a good agreement between the two indicators. Finally, we have good confidence in the ability of our method to identify events since the identified drought periods are in good agreement with similar studies found in the scientific literature (e.g. Hoerling et al., 2012; Spinoni et al., 2015; Raymon et al., 2016).

As we worked in clusters by type of extreme events, the bias on the intensity of the events should not affect the conclusions of the article. Additional information on the uncertainty of extreme events detected with the PLA method is added in the text and reference is made to the article by Guion et al. (2021). Please see Ln. 201 – 204 "$PLA_{T2m}$ was calculated based on 2m temperature observations (E-OBS data set) in Guion et al. (2021). Although the intensity of heatwaves was slightly overestimated with the Med-CORDEX simulations (+0.16°C for the mean bias and +0.50°C for the maximum bias), their temporal correlation with observations was high (R coefficient of about 0.9 over the whole Mediterranean)."

[Figure]

Figure E – Heatwaves detection based on the PLA method using the 2m above surface temperature from RegIPSL (leflt column) and from the E-OBS dataset (right column). The first row shows the value of the percentile 75, the second one the mean intensity of heatwaves and the third one the frequency of highly intense events. The left column is from Guion et al. (2021).

[Figure]

Figure F - Main characteristics of the identified heatwaves based on the Med CORDEX simulation and the E-OBS dataset using the PLA method for the 1979-2016 time period over the Western Mediterranean. The error bars on the middle panel (mean intensity) correspond to the standard deviation.

[Figure]

Figure G – Main characteristics of the identified heatwaves based on the Med-CORDEX simulation and the E-OBS dataset using the PLA method for the 1979-2016 time period over the Southern Italy and Balkans. The error bars on the middle panel (mean intensity) correspond to the standard deviation. Figure from Guion et al. (2021).

C) Model-OMI HCHO discrepancies look quite large, so are the model-obs isoprene discrepancies at one site. Can other satellite HCHO products be used to help determine the OMI HCHO uncertainty? Can more information be provided on data screening and the regridding approach (L146)? Also, note that modeled $NO_x$ uncertainty can contribute to the model-OMI HCHO discrepancies. Can in-situ $NO_2$ bias that has been noted be corrected according to Lamsal et al., or modeled $NO_2$ columns be compared with satellite data?

We focused the analysis on three sub-regions of Southwestern Europe (as defined in the Fig. 2, 5 and 8 of the manuscript). Absolute and relative values of HCHO presented in Table 3 of the manuscript, are computed cell by cell before spatial average. The main objective of using the OMI data in this study is to assess the variation of HCHO during droughts and heatwaves (compared to normal conditions) as a proxy of BVOCs emissions.

Therefore, as suggested by the reviewer, we compared HCHO variations to another OMI product (OMI-BIRA) over the 2005-2016 period. Please see Ln. 148 - 152 "In order to quantify the uncertainty on the HCHO anomalies obtained, the analysis has been performed using two products. We primarily use the OMHCHOd level 3 product (Chance, 2019), which provides HCHO total column with a spatial resolution of 0.1° × 0.1°. For comparison, the level 2 retrieval by the Belgian Institute for Space Aeronomy (BIRA) is also used (De Smedt et al., 2015), thereafter referred to as OMI-BIRA. Both products are included in the intercomparison conducted by Zhu et al. (2016).".

This helps to better determine OMI HCHO uncertainty in our results. Please see Ln. 568 - 572 "Finally, the observed variations of total HCHO over summers 2005-2006 and the three areas considered here were also computed with OMI-BIRA retrieval. During heatwaves, both show a significant increase: +31% for OMI-BIRA (and +15% for OMHCHOd). The increase is lower during droughts: +13% (and +3%). The variation becomes slightly negative during isolated droughts: -2% (and -6%). This comparison highlights the uncertainty in the satellite observations but also that the general behaviour is consistent in both retrievals and similar to what was obtained for 2012 using CHIMERE.". These ranges of values that allow to assess the uncertainty of the results are also present in the abstract and conclusions.

Finally, we have added information on the data screening and the regridding approach. Please see Ln. 153 – 156 "Only observations with a cloud fraction less than 0.3, a solar zenith angle less than 70° and a vertical column density within the range of $-0.8 \times 10^{15}$ and $7.6 \times 10^{15}$ mocelules/$cm^2$ are selected in order to minimize OMI row anomalies, following Zhu et al. (2017). For a suitable comparison with model simulations, HCHO data were regridded on the Med-CORDEX domain using a bilinear method.".

*Inaccurate language, statements needing more supporting evidence, typos, etc.*

A) Title and abstract: The title does not accurately reflect the paper contents which cover normal periods as well. The focused model processes could be specified.

We thank the reviewer for this comment. We agree and have decided to change the title as follows "Biogenic isoprene emissions, dry deposition velocity and surface ozone concentration during summer droughts, heatwaves and normal conditions in Southwestern Europe".

B) L3: met conditions not only modify photochemistry activity and vegetation states so this statement is not accurate. It also contradicts with L37 that met conditions also drive transport and the fact that met conditions drive biogenic emissions and dry deposition magnitude and variability that they study.

We agree that this part should be completed. Please see Ln. 2 – 5 "Meteorological conditions are key to understand the variability of $O_3$ concentration, especially during extreme weather events. In addition to modifying photochemical and atmospheric transport, droughts and heatwaves affect the state of vegetation and thus the biosphere-troposphere interactions that control atmospheric chemistry, namely biogenic emissions of precursors and gaz dry deposition.".

C) L4: "lack of interactions between biosphere and troposphere" is vague. It should be made clear whether you refer to the biosphere-atmosphere exchanges of water/energy that concern land/weather conditions, or the direct soil and vegetation controls on chemical processes such as biogenic emissions and dry deposition. This comment also applies to the "Interactions between the terrestrial biosphere and atmosphere" phrase in the title.

We refer to the interactions between soil/vegetation and the atmosphere that control the chemistry of the atmosphere. We have specified that. Please see Ln. 3 – 6 "In addition to modifying photochemical and atmospheric transport, droughts and heatwaves affect the state of vegetation and thus the biosphere-troposphere interactions that control atmospheric chemistry, namely biogenic emissions of precursors and gaz dry deposition. A major source of uncertainty and inaccuracy in the simulation of surface $O_3$ during droughts and heatwaves is the poor representation of such interactions".
We have also changed the title (comment A, Inaccurate language, etc., Reviewer #3).

D) L10-11: key factor of what? Isn't biomass decrease a result of drought/heat stresses as well? Please consider rewording this sentence.

The decrease in biomass is indeed a consequence of drought/heat stresses. However, we consider soil dryness and biomass decrease as two

drought/heatwave effects, since they are represented by two different parameters in the BVOC emissions and gaz dry gas deposition schemes.
We have changed this. Please see Ln. 12 - 14 "Our sensitivity analysis shows that the decrease in both soil moisture and biomass parameter during droughts induces a significant decrease in biogenic $C_5H_8$ emissions and $O_3$ dry deposition velocity. We find a larger impact induced by the variation of the soil moisture parameter. However, combined effects on surface $O_3$ remain limited.".

E) L17: "in agreement with HCHO satellite observations": this conclusion is questionable according to the results presented in the paper; "favorable" for what?

We refer mainly to the results presented in Table 3. We have rewritten the statement for satellite observations of HCHO. Please see Ln. 19 - 26 "However, we simulated a decrease of $C_5H_8$ emissions during isolated droughts (i.e. not accompanied by a heatwave), resulting in a non-significant difference of surface $O_3$ compared to normal conditions (from both observations and simulations). Despite a significant bias between the total columns of formaldehyde (HCHO, used as a proxy of biogenic emissions of volatile organic compounds) simulated by CHIMERE and observed by the Ozone Monitoring Instrument (Aura satellite), the satellite data confirm an average increase of HCHO (between +3 and +13% depending on the product used) over the three regions of interest (Balkans, Pô Valley and Central Spain) for all droughts and a decrease (between -2 and -6%) for isolated droughts, over summers 2005 to 2016.".
Furthermore, we refer to meteorological conditions that are favorable for photochemical activity.

F) L21-24 does not reflect the highlights of this studies and may be removed or rewritten.

As pointed out by the reviewer, it does not reflect the highlights of this studies. We have removed this point from the abstract.

G) L68: "because of" should be "partially because of"

We have modified this. Please see Ln. 68 – 70 "The variability of $O_3$ concentration is generally not well represented in chemistry-transport models (CTMs) compared to observations partially because of the lack of interactions between the meteorology, terrestrial biosphere and atmospheric chemistry (Wang et al., 2017).".

H) The connections between extreme weather and drought conditions in Section 1 are very nice and may be reorganized/sharpened with more citations from the land-atmosphere interaction communities being included, in terms of mechanisms, observation evidences and model capability/limitations, e.g., Miralles et al. https://doi.org/10.1029/2012GL053703; https://doi.org/10.1111/nyas.13912; Hirschi et al.

https://doi.org/10.1038/ngeo1032. The authors may want to note that land influences on atmosphere through evapotranspiration are included in their coupled WRF/ORCHIDEE systems and are not perturbed in this study.

We thank the reviewer for this comment. Land-atmosphere feedbacks are well taken into account in the coupled WRF-ORCHIDEE model (RegIPSL) and thus, these interactions are integrated in the dynamics of droughts and heatwaves detected by the PLA indicator. We have provided additional information on this subject in the introduction. Please see Ln. 80 – 83 "The development of droughts and heatwaves can be linked (Miralles et al., 2019). For example, through the soil moisture-temperature feedback, droughts can increase heatwave intensity due to a decrease in evapotranspiration and an increase in sensible heat (e.g. Zampieri et al., 2009). It is therefore important to integrate such interactions for the simulation of droughts and heatwaves.".

I) L294: Can this fitting function from Bonn et al. be written out here? For isoprene only?

As suggested by the reviewer, we have written the function of Bonn et al. (2019) in the manuscript for clarity (Ln. 331 - 332).
Bonn et al. (2019) also present a fitted function of an activity factor dependent on soil water availability for terpene species emissions. However, their responses change with the species considered. Due to the large uncertainties on terpene species, we did not include such a fitted function in our experiment.

J) L326: uncertainty in PBLH is not supported by any sort of analysis. I am assuming that the authors meant to refer to the nighttime poor performance of PBLH that has been seen in many models. Please confirm.

It is indeed the night-time representation of the Planetary Boundary Layer Height (PBLH) that is often misrepresented in WRF and difficult to evaluate because there are not many measurements for the boundary layer height. We have made this clear in the manuscript. Please see Ln. 608 – 613 "Meteorological conditions are critical for the $O_3$ budget especially during summer droughts and heatwaves. In addition to uncertainties in the modelling of precursor emissions and $O_3$ deposition (as mentioned above), differences between observations and simulations may also rely on meteorological uncertainties, such as the diurnal temperature cycle (see Sect. 4) and the Planetary Boundary Layer Height (PBLH). The night-time representation of the PBLH is often misrepresented in the WRF model (e.g. Chu et al., 2019).".

K) L332: change "norhtern" to "northern"

We have changed that. Please see Ln. 366 - 368 "The daily maximum $T_{2m}$ is overestimated in the Southern Mediterranean (up to 5°C) compared to observations while it is underestimated in the northern part (up to 5°C).".

L) L419-420: This sentence seems to be misplaced and should belong to the motivation of this study?

Since we have added a comparison of the ozone deposition flux at one station for the summer 2012 (see comment K of the minor issues from Reviewer #1), we think it is more interesting to leave it as a discussion of our results with the observational measurements and the literature review.

M) L587: "as" is the wrong word?

We have change this with "The second objective of this study was to quantify the variation of surface $O_3$ over the Southwestern Europe during agricultural droughts, combined or not with heatwaves. Those extreme weather events were identified based on the RegIPSL model using the PLA indicator." (Ln. 648 – 650).

N) 588: PLA should be defined on its first occurrence in the paper.

PLA is defined once in the abstract to be clear. It is then defined once in the body of the text, in Ln. 189 – 190 "Following the approach of Lhotka and Kyselý (2015), we computed the Percentile Limit Anomalies of 2m above surface temperature ($PLA_{T2m}$) for heatwave detection and of soil dryness ($PLA_{SD}$) for agricultural drought detection.".

O) Throughout the paper, many acronyms need to be defined on their first occurrences in the paper.

We agree that our paper has a large number of acronyms. We have already reduced the use of acronyms in the version submitted for the preprint. In addition, we have paid attention to this throughout the new version of the manuscript.

References in responses to comments:

Anav, A., Menut, L., Khvorostyanov, D., & Viovy, N. (2011). Impact of tropospheric ozone on the Euro-Mediterranean vegetation. *Global Change Biology*, *17*(7), 2342-2359.

Baumbach, L., Siegmund, J. F., Mittermeier, M., & Donner, R. V. (2017). Impacts of temperature extremes on European vegetation during the growing season. *Biogeosciences*, *14*(21), 4891-4903.

Bonn, B., Magh, R. K., Rombach, J., & Kreuzwieser, J. (2019). Biogenic isoprenoid emissions under drought stress: different responses for isoprene and terpenes. *Biogeosciences*, *16*(23), 4627-4645

Emberson, L. D., Simpson, D., Tuovinen, J. P., Ashmore, M. R., & Cambridge, H. M. (2000). *Towards a model of ozone deposition and stomatal uptake over Europe*. MSC-W.

Erisman, J. W., Van Pul, A., & Wyers, P. (1994). Parametrization of surface resistance for the quantification of atmospheric deposition of acidifying pollutants and ozone. *Atmospheric Environment*, *28*(16), 2595-2607.

Franz, M., Simpson, D., Arneth, A., & Zaehle, S. (2017). Development and evaluation of an ozone deposition scheme for coupling to a terrestrial biosphere model. *Biogeosciences*, *14*(1), 45-71.

Guion, A., Turquety, S., Polcher, J., Pennel, R., Bastin, S., & Arsouze, T. (2021). Droughts and heatwaves in the Western Mediterranean: impact on vegetation and wildfires using the coupled WRF-ORCHIDEE regional model (RegIPSL). *Climate Dynamics*, *58*(9), 2881-2903.

Hoerling, M., Eischeid, J., Perlwitz, J., Quan, X., Zhang, T., & Pegion, P. (2012). On the increased frequency of Mediterranean drought. *Journal of climate*, *25*(6), 2146-2161.

Müller, J. F., Stavrakou, T., Wallens, S., De Smedt, I., Van Roozendael, M., Potosnak, M. J., ... & Guenther, A. B. (2008). Global isoprene emissions estimated using MEGAN, ECMWF analyses and a detailed canopy environment model. *Atmospheric Chemistry and Physics*, *8*(5), 1329-1341.

Raymond, F., Ullmann, A., Camberlin, P., Drobinski, P., & Smith, C. C. (2016). Extreme dry spell detection and climatology over the Mediterranean Basin during the wet season. *Geophysical Research Letters*, *43*(13), 7196-7204.

Spinoni, J., Naumann, G., Vogt, J. V., & Barbosa, P. (2015). The biggest drought events in Europe from 1950 to 2012. *Journal of Hydrology: Regional Studies*, *3*, 509-524.

Wang, P., Liu, Y., Dai, J., Fu, X., Wang, X., Guenther, A., & Wang, T. (2021). Isoprene emissions response to drought and the impacts on ozone and SOA in China. *Journal of Geophysical Research: Atmospheres*, *126*(10), e2020JD033263.

Zhang, Q. J., Beekmann, M., Drewnick, F., Freutel, F., Schneider, J., Crippa, M., ... & Perrussel, O. (2013). Formation of organic aerosol in the Paris region during the MEGAPOLI summer campaign: evaluation of the volatility-basis-set approach within the CHIMERE model. *Atmospheric Chemistry and Physics*, *13*(11), 5767-5790.

Zhu, L., Jacob, D. J., Kim, P. S., Fisher, J. A., Yu, K., Travis, K. R., ... & Wolfe, G. M. (2016). Observing atmospheric formaldehyde (HCHO) from space: validation and intercomparison of six retrievals from four satellites (OMI, GOME2A, GOME2B, OMPS) with SEAC 4 RS aircraft observations over the southeast US. *Atmospheric Chemistry and Physics*, *16*(21), 13477-13490.

---

## Author Response (AR2)

Reply to the reviewer's comments on the manuscript "Biogenic isoprene emissions, dry deposition velocity and surface ozone concentration during summer droughts, heatwaves and normal conditions in Southwestern Europe".

Guion et al.

**Report #2**

The authors would like to thank the reviewers for their careful reading of the paper and for their comments that improved the quality of the manuscript. All comments have been addressed and a point-by-point answer is provided in the following (in blue after the corresponding comment). The line numbers given in response to comments correspond to the latest version submitted. Finally, modifications made in the new manuscript version are highlighted in the track-changes file provided by the authors.

*Suggestions for minor revisions:*

A. The abstract needs to be rewritten to provide quantitative information on the impacts of drought and heatwaves on biogenic isoprene emissions, dry deposition velocity, and surface ozone air pollution.
   Line 6 in the abstract: what is "gaz dry deposition"? Do you mean "gas dry deposition"?

We agree that the abstract can be improved by providing more quantified effects. We have rewritten the summary, in particular the part on cluster analysis.

We thank the reviewer for identifying the writing error. We made sure to replace the word "gaz" with "gas" everywhere.

B. Please label figure panels as a, b, c, … and explicitly describe each panel in the caption. Figures in the present form (e.g., Figure 2 and 5) are very difficult to follow.

As pointed out by the reviewer, we labeled the panels of the figures and updated the captions.

C. In the present manuscript, discussion on the variability of isoprene emissions, dry deposition, and ozone concentrations are based on episodes (mostly 2012). It would be more meaningful to examine inter-annual variability in 2012 (hot and dry), 2013, and 2014 (relatively cool and wet).

Figure 13 shows that the model has difficulty simulating the observed ozone inter-annual variability, particularly the observed ozone enhancements in the hot and dry summer of 2012. Does the model simulate increases in biogenic isoprene emissions and decreases in ozone deposition velocity in the summer of 2012 compared to 2014? In-depth discussion and analyses are needed for this section.

The analysis of the variation of $C_5H_8$ emissions, $O_3$ dry deposition, and $O_3$ concentrations by clusters of extreme weather events was well done for the summers of 2012, 2013 and 2014. It includes inter-annual variability.

The result section is divided into two parts. Firstly, we performed a sensitivity analysis of simulated biogenic $C_5H_8$ emissions, $O_3$ concentration and $O_3$ dry deposition to drought effects only for summer 2012. Secondly, we performed a cluster analysis of simulated biogenic $C_5H_8$ emissions, $O_3$ concentration and $O_3$ dry deposition over summers 2012-2014, including a comparison to $O_3$ over the same period. To allow more robust conclusions, we have extended the cluster analysis to the observations of $O_3$ for summers 2000-2016 and HCHO for summers 2005-2016. We made it clearer in the abstract and introduction of the section 5.2 "Statistical variation during droughts and heatwaves". Please see Ln. 512 – 515 "Clusters of droughts and heatwaves (isolated or combined) are constructed based on the $PLA_{T2m}$ and $PLA_{SD}$ indicators, allowing to analyze the statistical variation of simulated $C_5H_8$ emissions, $O_3$ stomatal conductance and $O_3$ surface concentration for summers 2012-2014. We performed the same analysis on observations of HCHO total column for summers 2005-2016 and $O_3$ for summers 2000-2016.".

Concerning the Figure 13, the number of exceedances is indeed generally smaller in the simulations (compared to the observations). This is due to the overall underestimated daily maximum in CHIMERE presented in the validation. However, some countries such as France or Austria present similar inter-annual patterns (Figure 13). Moreover, the distribution of exceedance days by extreme weather events over Europe is consistent with the observations (Table 4), as stated in Section 3 "Threshold level exceedance of $O_3$".